# Neutralizing antibodies after the third COVID-19 vaccination in healthcare workers with or without breakthrough infection

Arttu Reinholm [1] ✉, Sari Maljanen[1], Pinja Jalkanen[1], Eda Altan[1], Sisko Tauriainen [1], Milja Belik [1], Marika Skön [2], Anu Haveri [2], Pamela Österlund [2], Alina Iakubovskaia[3], Arja Pasternack [3], Rauno A. Naves[3], Olli Ritvos[3], Simo Miettinen[4], Hanni K. Häkkinen[5], Lauri Ivaska[6,7], Paula A. Tähtinen [6], Johanna Lempainen [1,6,8], Anu Kantele[4], Laura Kakkola [1,7,8], Ilkka Julkunen [1,7,8] & Pekka Kolehmainen [1] ✉

## Abstract

**Background** Vaccinations against the SARS-CoV-2 are still crucial in combating the ongoing pandemic that has caused more than 700 million infections and claimed almost 7 million lives in the past four years. Omicron (B.1.1.529) variants have incurred mutations that challenge the protection against infection and severe disease by the current vaccines, potentially compromising vaccination efforts.

**Methods** We analyzed serum samples taken up to 9 months post third dose from 432 healthcare workers. Enzyme-linked immunosorbent assays (ELISA) and microneutralization tests (MNT) were used to assess the prevalence of vaccine-induced neutralizing antibodies against various SARS-CoV-2 Omicron variants.

**Results** In this serological analysis we show that SARS-CoV-2 vaccine combinations of BNT162b2, mRNA-1273, and ChAdOx1 mount SARS-CoV-2 binding and neutralizing antibodies with similar kinetics, but with differing neutralization capabilities. The most recent Omicron variants, BQ.1.1 and XBB.1.5, show a significant increase in the ability to escape vaccine and infection-induced antibody responses. Breakthrough infections in thrice vaccinated adults were seen in over 50% of the vaccinees, resulting in a stronger antibody response than without infection.

**Conclusions** Different three-dose vaccine combinations seem to induce considerable levels of neutralizing antibodies against most SARS-CoV-2 variants. However, the ability of the newer variants BQ1.1 and XBB 1.5 to escape vaccine-induced neutralizing antibody responses underlines the importance of updating vaccines as new variants emerge.

## Plain language summary

During the COVID-19 pandemic, mass vaccination efforts against SARS-CoV-2 infection have provided effective protection against the virus and helped reduce the severity of symptoms in infected individuals. However, it is not well established whether the existing vaccines can provide the same protection against new and emerging SARS-CoV-2 variants that develop over time as the virus evolves. In this study, we tested combinations of three-dose COVID-19 vaccines given in random order to protect against all SARS-CoV-2 variants in circulation including the newest being Omicron variants. We demonstrate that more than half of the population who received the three-dose vaccine combinations were infected with SARS-CoV-2 Omicron variants after receiving the last vaccine dose. These findings indicate the need to develop new vaccine candidates against emerging SARS-CoV-2 variants.

[1]Institute of Biomedicine, University of Turku, Turku, Finland. [2]Finnish Institute for Health and Welfare, Helsinki, Finland. [3]Department of Physiology, Medicum, University of Helsinki, Helsinki, Finland. [4]Department of Infectious Diseases, Meilahti Vaccination Research Center, MeVac, Helsinki, University Hospital and University of Helsinki, Helsinki, Finland. [5]Department of Virology, University of Helsinki, Helsinki, Finland. [6]Department of Paediatrics and Adolescent Medicine, Turku University Hospital and University of Turku, Turku, Finland. [7]InFlames Research Flagship Center, University of Turku, Turku, Finland. [8]Clinical Microbiology, Turku University Hospital, Turku, Finland. ✉e-mail: arttu.j.reinholm@utu.fi; pekka.j.kolehmainen@utu.fi

Severe acute respiratory syndrome coronavirus −2 (SARS-CoV-2) is a coronavirus from the *betacoronaviridae* genus that has been circulating around the globe for over four years. Vaccination remains an important tool in restricting the spread of the virus and reducing the number of severe COVID-19 cases. The growing number of breakthrough infections affects the immunity against COVID-19 but also raises concern about the efficacy of the COVID-19 vaccines against the newer variants-of-concern (VOC), namely the Omicron variants. The first Omicron variant BA.1, was first identified in mid-November 2021 in South Africa, after which other Omicron variants rapidly sprouted around South Africa[1]. Omicron variants BQ.1.1 and XBB.1.5 are the most dominant variants in many countries, including Finland, where a prevalence of 95–98% was seen in March 2023[2]. Omicron BA.1 has 62 amino acid changes compared to the ancestral Wuhan strain, half of which are concentrated in the spike (S) protein. S protein facilitates receptor binding and virus-host membrane fusion, and cumulated amino acid changes allow BA.1 to evade host antibody responses more efficiently while retaining its receptor-binding affinity[3]. Omicron BA.4 and BA.5 are closely related to BA.2 and thus, more distinct from the BA.1 variant; however, they still possess similar antibody evasion capabilities as BA.1[4]. Recent Omicron variants, BQ.1.1 (BA.5 subvariant) and XBB.1.5 (BA.2 subvariant), have acquired mutations that further increase their immune evasion capabilities while retaining receptor-binding affinity[5].

The first three COVID-19 vaccines used during the pandemic in Finland were mRNA-based BNT162b2/Comirnaty (Pfizer-BioNTech) and mRNA-1273/Spikevax (Moderna), and adenovirus-vector-based ChAdOx1-S/Vaxzevria (AstraZeneca-Oxford), all of which elicit antibody responses against the S protein[6,7]. In this study, we systematically compared the antibody response induced by combinations of these vaccines. In addition, we carried out an extensive analysis of vaccine-induced antibody responses, spike-specific and neutralizing antibodies against the recently circulated Omicron variants, including those of BQ.1.1 and XBB.1.5. We also compared the antibody responses in vaccinated participants without and with breakthrough infections.

Our results indicate that three-dose vaccination combinations used in Finland and breakthrough infections induce significant levels of neutralizing antibodies against older SARS-CoV-2 variants, but are less effective at inducing neutralizing antibodies against the newer BQ.1.1 and XBB.1.5 variants.

## Methods

### Study population
A study cohort of 432 healthcare workers (HCWs) from University Hospitals of Turku and Helsinki received two sequential doses of COVID-19 vaccine with two dosing intervals: a short, three-week interval ($n = 230$) or a long, 12-week interval ($n = 202$, Table 1) followed by a third dose (received by 417 of 432 vaccinees, 97% of the vaccinees). HCWs with a short dose interval were vaccinated with two doses of BNT162b2 and a third dose of BNT162b2 ($n = 98$) or mRNA-1273 ($n = 132$) vaccine 9 months (range 6.2–12.4) after the second dose. HCWs with a long dose interval received two doses of BNT162b2 ($n = 62$) or mRNA-1273 ($n = 68$) vaccines or a combination of ChAdOx1 and BNT162b2 or mRNA-1273 ($n = 72$) vaccines before a third dose of BNT162b2 ($n = 97$) or mRNA-1273 ($n = 90$) vaccine six (2.7–9.8) months after the second dose. In a short dose interval group, we analyzed samples preceding and following the third vaccine dose (Fig. 1a). In a long interval group, serum samples were collected before the vaccinations and in regular intervals after each vaccine dose (Fig. 2a). The vaccinees filled out symptom questionnaires and information regarding SARS-CoV-2 antigen and RT-qPCR test results at every sample collection. Supplementary Table 1 summarizes the number of samples at each time point for each vaccine group.

### Enzyme immune assay
The levels of SARS-CoV-2 S1 and N-specific IgG antibodies in the serum samples were measured by an in-house EIA as described previously[8]. Briefly, purified recombinant SARS-CoV-2 S1 (3.5 µg/ml) and N (2.0 µg/ml) antigens diluted in PBS were absorbed to 96-well immunoplates (Thermo Scientific). Serum dilutions of 1:300 and 1:1000 were prepared into EIA-buffer (5% swine serum in 0.05% Tween-20 in PBS) for detection of N-binding and S1-binding IgG antibodies, respectively, and quantified as the absorbance at a 450 nm wavelength. Measured OD (optical density) value for each sample was converted into EIA units based on the OD values of a known positive (100 EIA units for N and 115 EIA units for S1) and a negative control (0 EIA units). Cut-off values for seropositivity were calculated in two steps. First a preliminary cut-off was calculated as the average of EIA unit values from pre-vaccination samples plus three times the standard deviation (excluding participants with a prior PCR-confirmed SARS-CoV-2 infection). Then the

**Table 1 | Demographics and breakthrough infection rates of COVID-19 vaccinated HCWs ($n = 432$)**

| | Short dose interval 2x BNT162b2 | Long dose interval 2x COVID-19 vaccine | Long dose interval 2x COVID-19 vaccine ($n = 202$) | | |
| --- | --- | --- | --- | --- | --- |
| | | | 2x BNT162b2 | 2x mRNA-1273 | ChAdOx + 1x BNT162b2/ mRNA-1273 |
| N | 230 | 202 | 62 | 68 | 72 |
| Female (%) | 201 (87.4%) | 184 (91.1%) | 56 (90.3%) | 63 (92.6%) | 65 (90.3%) |
| Male (%) | 29 (12.7%) | 18 (8.9%) | 6 (9.7%) | 5 (7.4%) | 7 (9.7%) |
| Age in years | | | | | |
| Mean | 43 | 46 | 46 | 44 | 47 |
| Median | 41 | 46 | 49 | 43 | 48 |
| Range | 19–65 | 22–67 | 22–66 | 25–64 | 23–67 |
| Mean time between vaccine doses (range) | | | | | |
| Between 1st and 2nd dose in weeks | 3.0 (2.6–4.7) | 12.2 (8.0–27.7) | 11.7 (8.0–15.0) | 12.2 (11.9–21.1) | 12.5 (10.9–27.7) |
| Between 2nd and 3rd dose in months | 8.6 (6.2–12.4) | 6.1 (2.7–9.8) | 6.2 (2.7-9.8) | 5.6 (4.7–7.0) | 6.6 (4.2–8.5) |
| PCR test or antigen test confirmed infection | | | | | |
| Prior to vaccination | 6 | 15 | 6 (10.0%) | 1 (1.5%) | 8 (11.1%) |
| Between the second and third vaccine dose | 0 | 10 (4.9%) | 5 (8.1%) | 2 (2.9%) | 3 (4.2%) |
| After three vaccine doses | 98 (42.6%) | 80 (39.6%) | 25 (40.3%) | 30 (44.1%) | 25 (34.7%) |
| Total | 104 (45.2%) | 105 (52.0%) | 36 (58.1%) | 33 (48.5%) | 36 (50.0%) |

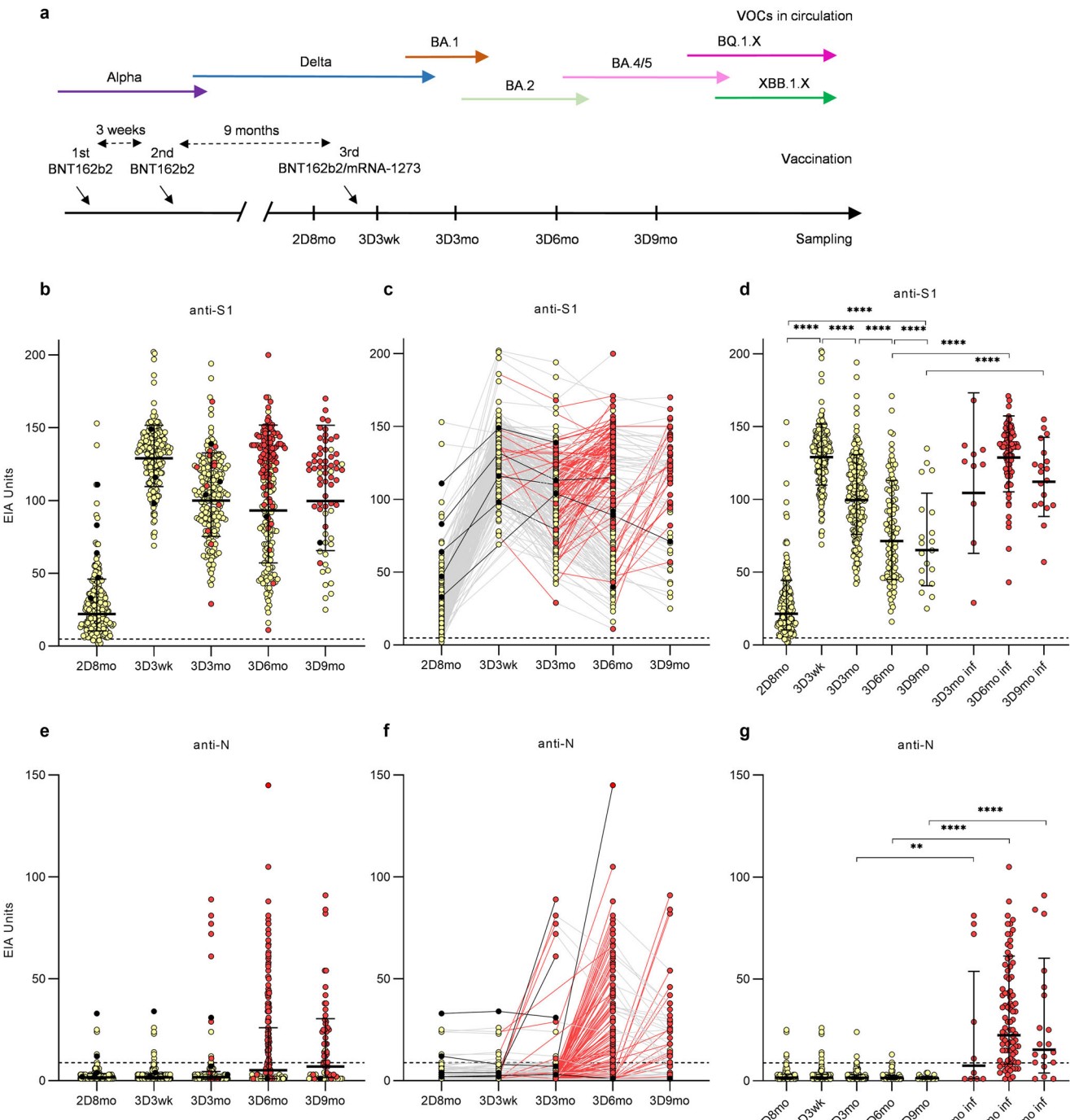

**Fig. 1 | Timeline of vaccination and serum sampling, and SARS-CoV-2 S1-specific and N-specific IgG antibody responses of short-interval vaccinated healthcare workers up to 9 months after the third vaccine dose. a** Serum samples were collected from 230 HCWs who received two BNT162b2 vaccine doses with a short 3-week interval and a third dose BNT162b2 or mRNA-1273 9 months after the second dose. Sera were collected 8 months after the second vaccine dose (2D8mo; *n* = 224) preceding the third dose, and 3 weeks (3D3wk, *n* = 222), 3 months (3D3mo, *n* = 217), 6 months (3D6mo, *n* = 206), and nine months (3D9mo, *n* = 66) after the

third vaccine dose. EIA was used to measure **b–d** SARS-CoV-2 S1-specific and **e–g** N-specific IgG antibody levels. Red dots and lines indicate antibody levels after a SARS-CoV-2 breakthrough infection. Individuals who had a SARS-CoV-2 infection before vaccination are marked with black dots and lines. **d, g** present the antibody levels of infected individuals separated from non-infected. Geometric means and geometric standard deviations of antibody levels are shown. Dashed lines indicate the cut-off values for seropositivity. Time points before 3D6mo have been partially analyzed in previous studies[7,8,10].

final cut-off values were calculated as the average of EIA unit values plus three times the standard deviation using the values, which remained below the preliminary cut-off. In assessing breakthrough infections, cut-off values were used to gauge if the pre- and post-breakthrough samples differed substantially. If the difference was greater than the cut-off value, it was determined that a breakthrough infection had occurred.

## SARS-CoV-2 strains

SARS-CoV-2 strains were isolated and grown from SARS-CoV-2 PCR-positive nasopharyngeal samples in Finland: FIN25-20 (Pango lineage B.1, D614G strain, GenBank ID MW717675.1 and GISAID ID EPI_ISL_412971), FIN55-21 (Omicron BA.1 variant, EPI_ISL_8768822.2), FIN58-22 (Omicron BA.2 variant, OP199045 and EPI_ISL_9695067),

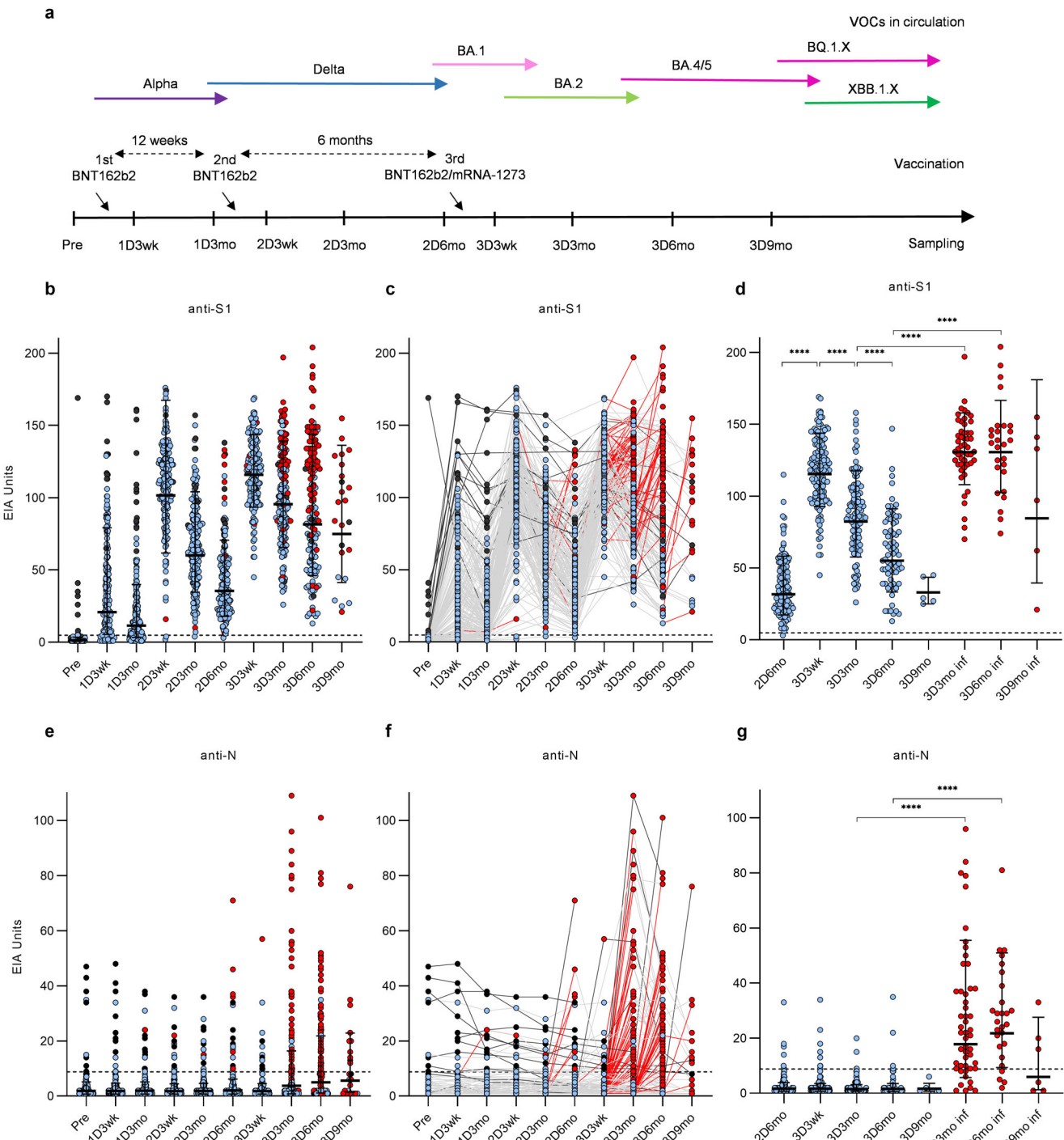

**Fig. 2 | Timeline of vaccination and serum sampling, and SARS-CoV-2 S1-specific and N-specific IgG antibody responses of long, 12wk-interval vaccinated healthcare workers up to 9 months after the third vaccine dose. a** Serum samples were collected from 202 HCWs who received two COVID-19 vaccine doses (2x BNT162b2, 2x mRNA-1273, or ChAdOx1 + BNT162b2/mRNA-1273) with a long 12-week interval (8–27 weeks) and a third dose of BNT162b2 or mRNA-1273 6 months (2.7–9.8) after the second dose. Sera were collected from the vaccinees before vaccination (Pre, $n = 119$), 3 weeks (1D3wk, $n = 184$) and 3 months (1D3mo, $n = 174$) after the first dose (1D), 3 weeks (2D3wk, $n = 181$), 3 months (2D3mo, $n = 187$), and 6 months (2D6mo, $n = 150$) after the second dose (2D), and 3 weeks (3D3wk, $n = 164$), 3 months (3D3mo, $n = 176$), six months (3D6mo, $n = 156$), and

nine months (3D6mo, $n = 24$) after the third dose (3D). **b–d** SARS-CoV-2 S1-specific and **e–g** N-specific IgG antibody levels were measured from the sera with EIA. Red dots and lines indicate antibody levels after a SARS-CoV-2 breakthrough infection. Antibody levels of individuals who had a SARS-CoV-2 infection before vaccination are marked with black dots and lines. **d** and **g** show separately the antibody levels from uninfected individuals before and after the third vaccine dose and from those infected after the third vaccine dose. Geometric means and geometric standard deviations of antibody levels are shown. Dashed lines indicate the cutoff values for seropositivity. Time points before 3D3wk have been partially analyzed in a previous study[7].

FIN61-22 (Omicron BA.5 variant, OP199047 and EPI_ISL_13118918), FIN65-22 (Omicron BQ.1.1 variant, OQ411064 and EPI_ISL_15762173), and FIN69-22 (Omicron XBB.1.5 variant, OQ509907 and EPI_ISL_16526646). VeroE6 (for FIN25-20) or VeroE6-TMPRSS2-H10 cells (for FIN55-21, FIN58-22, FIN61-22, FIN65-22, and FIN69-22) were used to isolate the viruses and they were further passaged in VeroE6-TMPRSS2-H10 cells[9] in DMEM supplemented with 2% fetal calf serum, 2 mM L-glutamine, and 2 mM penicillin–streptomycin. Virus stock titers were determined by median Tissue Culture Infectious Dose ($TCID_{50}$) assay as described before[7,8].

## Microneutralization tests

Neutralizing antibody titers in the sera were determined with a micro-neutralization test (MNT) as shown before[7,8]. Briefly, starting from a 1:5 dilution, a two-fold dilution series was prepared for each serum sample into DMEM (supplemented with 2% fetal calf sera, 2 mM penicillin-strepto-mycin, 2 mM L-glutamine) and 50 $TCID_{50}$ of virus was added to the dilutions resulting in dilutions of serum from 1:10 up to 1:5120 (D614G) or 1:1280 (Omicron variants). Virus-serum dilution mixture was incubated at +37 °C and 5% $CO_2$ for 1 h before adding VeroE6-TMPRSS2-H10 cells (50,000 cells per well). After incubation for 4 days at +37 °C the cells were fixed with 4% formaldehyde, stained with crystal violet, and visualized for cell death. Reciprocal of the serum dilution able to inhibit 50% of cell death was considered as the neutralization titer. Serum dilution inhibiting 50% of cell death at a dilution of 1:10 or above was considered positive for neutralizing antibodies. Positive control was included in every plate for MNT.

## Statistics and reproducibility

All data was collected and arranged in Excel 2016 (Microsoft 365). GraphPad Prism (version 8.4.2) was utilized to analyze all collected data and to create graphs. Statistical analysis was conducted using a two-sided Wilcoxon signed-rank test for paired samples when applicable. The two-sided Mann–Whitney $U$ test or two-sided Kruskal–Wallis test followed by Dunn's multiple comparisons was used for groups with non-paired samples. $P$ values $< 0.05$ were considered statistically significant. Star symbols (*) in figures indicate $p$ values as follows: *$p$ value $< 0.05$, **$p$ value $< 0.002$, ***$p$ value $< 0.0002$, and ****$p$ value $< 0.0001$. Correlations were analyzed with Spearman's correlation test and the correlations are presented with a regression line fixed for the logarithmic axis. Protein structure models indicating the amino acid changes of the VOCs in relation to the original Wuhan isolate were created with UCSF Chimera (version 1.15) using the PDB ID 7WK2 as the core model for spike proteins. Two technical replicates were measured for serum samples in EIA and MNT to analyze the binding and neutralizing antibody levels.

## Ethical statement

HCWs from Turku University Hospital (Turku, Finland) (Southwest Finland health district ethical permission ETMK 19/1801/2020, EudraCT 2021-004419-14) and Helsinki University Hospital (Helsinki, Finland; Helsinki-Uusimaa health district ethical permission HUS/1238/2020, EudraCT 2021-004016-26) were recruited in the study prior to receiving COVID-19 vaccines as part of occupational healthcare. Written informed consents were collected from all participants before the collection of the first study sample.

## Reporting summary

Further information on research design is available in the Nature Portfolio Reporting Summary linked to this article.

## Results

### Demographics of and reported breakthrough infections in the study population

This study continued the follow-up of humoral antibody responses of healthcare workers (HCWs) who received two BNT162b2 vaccine doses with short three-week interval ($n = 230$, short group) or two doses of BNT162b2, mRNA-1273, or ChAdOx1 plus BNT162b2/ mRNA-1273 with a long 12-week ($n = 202$, long group) interval before the third vaccine dose. We have previously analyzed this cohort up to 3 months after the third vaccine dose[7,8,10]. The novel data obtained in this study is the analysis and follow-up of serum samples up to 9 months after the 3rd vaccine dose. In Finland, COVID-19 vaccinations were first carried out with a short three-week interval, and later in the spring of 2021 the interval was lengthened to 12 weeks. Thus, the vaccinees in the short interval group received their third dose before those in the long interval group. Our analysis included serum samples collected before the first vaccination and before the third vaccination up to nine months after the third vaccine dose (short and long interval groups) (Figs. 1a and 2a). The age range of the participants at the time of the first vaccine dose ranged from 20 to 65 years (mean 42.7 and median 41.2 years) in the short interval group and 22–67 years (mean 45.7 and median 46.3 years) in the long interval group (Table 1). Eighty-nine percent of the study participants were females.

Six of the vaccinees in the short interval group and fifteen in the long interval group had had a SARS-CoV-2 infection (confirmed by PCR) before the first vaccination. The number of PCR/antigen test-confirmed SARS-CoV-2 infections increased in the study population (and in Finland) after the emergence of the Omicron variants. The majority of the vaccinees had received their third dose by the end of 2021, and thus, the breakthrough infections occurred mostly after the third vaccine dose: HCWs in the short interval group reported 98 infections after the third dose and HCWs in the long interval group reported 10 infections after two doses and 80 after three vaccine doses (Table 1).

### COVID-19 mRNA-vaccine-induced antibody responses of HCWs vaccinated with a short dose interval up to nine months after the third dose

To estimate the changes in humoral immunity after vaccination with three doses of COVID-19 mRNA vaccines and potential exposure to circulating SARS-CoV-2 variants, we analyzed the changes in SARS-CoV-2 spike protein subunit 1 (S1) and nucleoprotein (N) specific IgG antibody levels in the sera of HCWs in the short interval group (Fig. 1, the timeline of the vaccinations, serum collections and circulating variants is shown in Fig. 1a). The third vaccination induced strong S1-specific IgG responses, which peaked at 3 weeks post the vaccination and decreased thereafter (Geometric mean of antibody levels 128.9 EIA units at 3D3wk, 99.96 EIA units at 3D3mo, 93.19 EIA units at 3D6mo, and 99.73 EIA units at 3D9mo; Fig. 1b). The reported (PCR or antigen test confirmed) or serologically observable (increase in S1-specific or N-specific antibodies greater than the cut-off, 4.8 or 8.8 EIA units, respectively) SARS-CoV-2 infections increased the S1-specific and N-specific antibody levels (Fig. 1b, e). Follow-up of the vaccinees showed that the majority of infected vaccinees had an increase in anti-S1 and anti-N antibodies after the time period of a reported infection (red lines in Fig. 1c, f).

Five vaccinees in the short interval group had SARS-CoV-2 N-specific antibodies prior to vaccination[8] (Fig. 1e): Two of these had had a PCR test confirmed SARS-CoV-2 infection before the vaccination while the other three had elevated but stable anti-N antibody levels without a sign of antibody decline. Grouping of the vaccinees based on their infection status (Fig. 1d, g) showed that the anti-S1 and anti-N levels were significantly different between HCWs with or without a breakthrough infection at 6 months after the third vaccine dose (non-infected 72 EIA units vs. infected 112 EIA units for S1, $p < 0.0001$ and 1.4 EIA units vs. 22 EIA units for $N$, $p < 0.0001$) after the time point of three months after the third vaccine dose. In some vaccinees, the infection-induced antibody response was delayed and, therefore, was detectable only in the later time points, while 13 infected vaccinees showed no or little changes in S1-/or N-specific antibodies (Fig. 1c, f).

### Antibody responses of HCWs vaccinated with a long dose interval up to 9 months after the third dose

HCWs with a long vaccine interval received two doses of BNT162b2, mRNA-1273, or a combination of ChAdOx1 and BNT162b2/mRNA-1273

as the first two doses (Table 1). Sequential serum samples were collected before the vaccination and at regular intervals after each vaccine dose and analyzed for S1- and N-specific IgG antibody levels (Fig. 2, the timeline of the vaccinations, serum collections and circulating variants is shown in Fig. 2a). The first vaccine dose induced a wide range of S1-specific antibody levels, and substantially higher levels in previously infected HCWs (Fig. 2b, c, black dots). While the second and the third doses increased S1-specific antibody levels higher, follow-up of the antibody levels of each vaccinee showed increases also before the third dose and after the first time point post the third vaccine dose (Fig. 2c). A vast majority of these increases coincided with a reported SARS-CoV-2 infection (PCR test or antigen test confirmed) preceding the sampling and most of these HCWs also had an increase in N-specific antibodies at the same time (Fig. 2e, f).

A separate analysis of antibody responses in individuals that had or had not contracted an infection (PCR or antigen test confirmed, or serologically observable) showed a decrease (Geometric mean 115.4 EIA units at 3D3wk vs 32.94 EIA units at 3D9mo) in anti-S1 antibody levels prior to the next vaccine dose in case there was no infection ($p < 0.0001$; Fig. 2d). A SARS-CoV-2 breakthrough infection increased the S1-specific antibody levels when responses of infected and uninfected were compared three and six months, respectively, after the third dose (Geometric mean 82.30 EIA units at 3D3mo vs 130.6 EIA units at 3D3mo infected, 55.02 EIA units at 3D6mo vs 130.6 EIA units at 3D6mo infected) $p < 0.0001$; Fig. 2d). Also, the N-specific antibody levels were induced by an infection, although the N-specific response was absent or low in 25 of the infected vaccinees (Fig. 2e–g). A decline in N-specific antibodies was observed in the sequential samples from participants infected pre-vaccination or post-vaccination while a few vaccinees had relatively high basal N-antibody levels without an indication of a decline in antibody (blue dots above cutoff value, Fig. 2f).

### Agreement of serology and COVID-19 PCR or antigen test
Of the 412 HCWs who had serum samples available after the third COVID-19 vaccine dose, 44% (182/412) reported a positive COVID-19 test (PCR or antigen) post third vaccine dose. Furthermore, 95% (173/182) of the COVID-19 test-positive HCWs showed a serological indication of infection, defined as a diagnostic increase in either S1- or N-specific antibodies (increase greater than cut-off, 4.8 EIA units for S1 after the time point of 3D3wk, and 8.8 EIA units for N after the third dose; Table 2). Of the COVID-19 test negative (or untested) HCWs, 13% (30/246) had a serological indication of infection by S1- or N-specific antibody levels, making the proportion of HCWs with a breakthrough infection, confirmed either by COVID-19 test or serology, after three vaccine doses 51% (212/412). Of note, measuring only S1 or N-specific antibodies was less sensitive at detecting infected individuals, since only 64% (117/182) of the COVID-19 test-positive HCWs had an increase in both anti-S1 and anti-N antibodies. Thus, serological detection of past infection was most accurate when IgG antibodies for both S1 and N antigens were positive.

We also examined the antibody levels of HCWs and compared those to their COVID-19 test and serological status. We found that infection soon after a COVID-19 vaccination was more likely to lead into lack of serological indication of infection. Seventeen per cent (10/60) of the HCWs who had a short interval between the third COVID-19 vaccine dose and a positive COVID-19 test (vaccination less than three months before a positive COVID-19 test, Supplementary Fig. 1) had no serological indication of an infection. Only 5% of the HCWs with a longer vaccine interval between

COVID-19 vaccination and infection lacked serological indication of an infection (vaccination three to six months before a positive COVID-19 test, Supplementary Fig. 2).

### Comparison of the third vaccine dose-induced IgG antibody responses by BNT162b2 and mRNA-1273
To determine the comparative efficiency of BNT162b2 and mRNA-1273 as the third dose, S1-specific antibody responses were analyzed after the third vaccine dose in uninfected HCWs. In both short and long-interval groups, the vaccine-induced antibody responses had similar kinetics, with the S1 antibody level peaking at 3 weeks after vaccination, followed on an average by a 25 EIA unit decline in subsequent samples (Fig. 3a, b). The mRNA-1273 vaccine as the third dose provided significantly higher S1-specific antibody responses in both interval groups ($p < 0.0423$ in the short interval group and $p < 0.0300$ in the long interval group). The difference between the groups receiving BNT162b2 or mRNA-1273 as the third dose in the long interval group was observable already prior to the third dose (a result of different vaccine combinations before the third dose, different vaccine combination groups marked with different colors in Fig. 3b), and the difference in antibody levels remained observable also after the third dose (Fig. 3b).

### Age-related antibody responses to three doses of COVID-19 vaccines
Our study included HCWs aged 19 to 67 years at the time of the first vaccine dose. To examine whether age affects humoral immune responses induced by COVID-19 vaccines, we analyzed anti-S1 IgG antibody levels in relation to age from uninfected HCWs (Supplementary Fig. 3). Some decrease in antibody levels was observed by increasing age, but a higher age did not prevent the induction of high antibody levels. Although the oldest age group, 55–67-year-olds, had, on average, the lowest antibody levels at 3 and 6 months after the third vaccine dose the antibody levels were relatively equal between all age groups (Supplementary Fig. 4).

### Neutralizing antibodies against Omicron BA.1, BA.2, BA.5, BQ.1, and XBB.1.5 variants up to six months after the third vaccine dose
Neutralizing capacity of the sera against SARS-CoV-2 variants was first examined in the short interval group by randomly selecting a subset of 41 HCWs (no prior PCR-confirmed SARS-CoV-2 infection). MNT was used to analyze in vitro neutralizing antibody titers against the ancestral D614G variant and the five recent SARS-CoV-2 Omicron variants, BA.1, BA.2, BA.5, BQ.1.1, and XBB.1.5 in serum samples collected at six months after the second vaccine dose (2D) and 3 weeks (3wk), three months (3mo), and six months (6mo) after the third booster dose (3D). (Fig. 4a–l).

Six months after the second vaccine dose, 95% of the vaccinees (39/41) neutralized the D614G and 44% (18/41), 76% (31/41), 90% (37/41), 5% (2/41), and 0% (0/41) neutralized Omicron BA.1, BA.2, BA.5, BQ.1.1, and XBB.1.5 variants, respectively (Fig. 4a–f). The third vaccine dose increased neutralizing antibodies against all variants, although the levels against BQ.1.1 and XBB.1.5 variants remained lowest. Despite the gradual decrease in the levels of neutralizing antibodies after the third dose, the geometric mean titers (GMT) were 4.4–7.2x higher six months after the third dose in comparison to 6 months after the second dose, (80 vs 364 for D614G, 9 vs 47 for Omicron BA.1, 24 v. 173 for BA.2, and 26 vs 125 for BA.5; $p < 0.0001$ for each pair). For Omicron BQ.1.1 and XBB.1.5 variants, the GMTs were 2.0x and 2.4x higher (5 vs. 12 for Omicron BQ.1.1, $p = 0.0003$, and 5 vs 10 for

**Table 2 | Agreement of COVID-19 PCR or antigen test-positivity and serology-confirmed infection in HCWs after three vaccine doses (n = 412)**

| | COVID-19 PCR/antigen test positive (n = 182) | COVID-19 test negative or not tested (n = 230) |
|---|---|---|
| Increase in antibodies[a] (%) | 173 (95%) | 30 (13%) |
| No increase in antibodies[a] (%) | 9 (5%) | 200 (87%) |

[a]Anti-SARS-CoV-2 S1 or N antibodies.

**a**

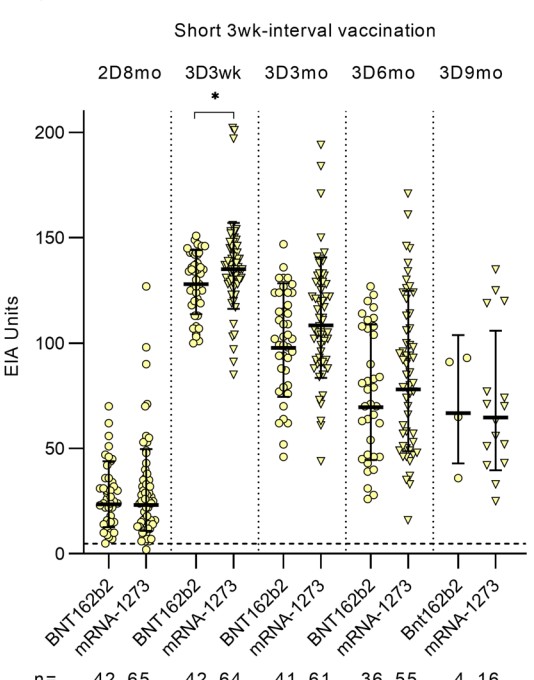

**b**

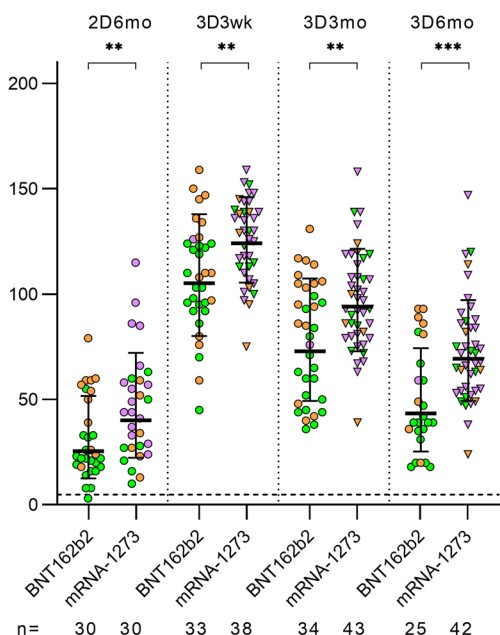

**Fig. 3 | Comparison of antibody responses induced by the third COVID-19 vaccination with BNT162b2 or mRNA-1273.** SARS-CoV-2 S1-specific IgG antibody responses induced by Bnt162b2 or mRNA-1273 (triangles) as the third vaccine dose were compared in the sera of uninfected vaccinees who received **a** two doses of Bnt162b2 with a short, three-week dose interval or **b** two doses of Bnt162b2 (orange), mRNA-1273 (violet), or ChAdOx1 + Bnt162b2/mRNA-1273 (green) with a long dose interval before the third vaccine dose. SARS-CoV-2 S1-specific IgG antibody responses were analyzed by EIA in serum samples collected before the third dose (2D6mo or 2D8mo) and after the third vaccine dose (3D3wk, 3D3mo, 3D6mo, and 3D9mo). Geometric means geometric standard deviations of antibody levels, and number of samples in each time point are shown. Dashed lines indicate the cut-off values for seropositivity.

Omicron XBB.1.5; $p = 0.0001$). Six months after the third dose only two samples had titers below the detection limit for Omicron BA.1, one sample for BA.2, 12 samples for BQ.1.1 and 17 samples for XBB.1.5. The results suggest that the neutralization efficiency of the induced antibodies is still reasonably high against the earlier Omicron variants, but is strongly reduced against BQ.1.1 and XBB.1.5 variants, with the number of samples below the detection limit increasing at the 3D6mo time point.

Twelve of the 41 vaccinees reported a PCR-confirmed SARS-CoV-2 breakthrough infection between the sampling of three and six months after the third dose. The breakthrough infection increased the levels of neutralizing antibodies, and the GMTs were 6–25x higher compared to non-infected participants (364 in non-infected vs. 2239 in infected at 3D6mo for D614G, 47 vs. 644 for Omicron BA.1, 173 vs. 997 for BA.2, and 125 vs. 910 for BA.5, 5 vs.126 for BQ.1.1, and 5 vs. 109 for XBB.1.5; $p = 0.0002$ for D614G and Omicron BA.2, $p < 0.0001$ for Omicron BA.1 and BA.5, and $p < x$ for Omicron BQ.1.1 and XBB.1.5). Of note, saturation in the measurement of neutralizing antibodies affected the titers for D614G, Omicron BA.2 and BA.5, and thus the difference in GMTs in infected vs. non-infected vaccines for these variants is likely somewhat higher. The BQ.1.1 is a subvariant of BA.5 and XBB.1.5 a subvariant of BA.2 with both having acquired mutations around the S1-RBD domain in S protein, the R346T, N460K, and F486V/P as the most notable amino acid changes[5,11,12] (Fig. 4m–q). Lower neutralizing capacity against BQ.1.1 and XBB.1.5 indicates that the acquired mutations are favorable for SARS-CoV-2 in avoiding neutralizing antibodies.

### Comparison of neutralizing antibody responses induced by different three-dose combinations of COVID-19 vaccines

To study differences in neutralizing antibody titers elicited by different vaccine combinations and vaccine dose intervals against D614G and Omicron BA.1. BA.2, BA.5, BQ.1.1, and XBB.1.5 sera from a representative number of HCWs (with two BNT162b2 with a short ($n = 41$) or with a long vaccine dose interval ($n = 35$), two mRNA-1273 ($n = 31$), or ChAdOx1 and BNT162b2 or mRNA-1273 ($n = 45$) before the third dose of BNT162b2 or mRNA-1273) were analyzed with MNT (Fig. 5). At 6 months after the second vaccine dose, the majority of vaccinees in each vaccine combination group had neutralizing antibodies against D614G variant and Omicron BA.2 and BA.5, whereas all vaccine combination groups had lower or undetectable levels of neutralizing antibodies against Omicron BA.1, BQ.1.1, and XBB.1.5. Interestingly, 2 x mRNA-1273 induced higher neutralizing antibody titers at 2D6mo against all variants, before the administration of the third vaccine dose. The third vaccine dose increased neutralizing antibody titers in all vaccine groups resulting in neutralizing antibody titers above the detection limit against all Omicron variants, leaving only four vaccinees below the detection limit against BQ.1.1 and ten against XBB.1.5. Interestingly, in Omicron variants 3 weeks after the third dose, the short interval 2 x BNT162b2 vaccine group had the highest GMT, even 1.1–1.9x higher than in the 2 x mRNA-1273 group (Fig. 5).

Three months after the third vaccine dose, the decrease in neutralizing antibody titers against D614G, and the Omicron variants was similar in all vaccine combination groups, and the neutralizing titers remained slightly higher in the 2 x BNT162b2 and 2 x mRNA-1273 vaccine groups than in the other groups. The difference in neutralization of Omicron BA.1 and XBB.1.5 was significant when the titers in short 2 x BNT162b2 group were compared to titers in ChAsOx1 + BNT162b2/mRNA-1273 group (GMT 40 vs. 99 against BA.1, $p = 0.0014$; GMT 8 vs. 16 against XBB.1.5, $p = 0.0024$) and 2x mRNA-1273 group (GMT 8 vs. 16 against XBB.1.5, $p = 0.040$).

Between the sampling of 3 weeks and three months after the third dose, 24 of the HCWs with the long vaccine interval reported a COVID-19 test-positive SARS-CoV-2 breakthrough infection (red dots in Fig. 5). The neutralizing antibody titers against the four variants were significantly higher in infected than in non-infected HCWs. Only one HCW with three vaccine doses and an infection had neutralizing antibody titers against Omicron XBB.1.5 below the detection limit. Altogether, these results

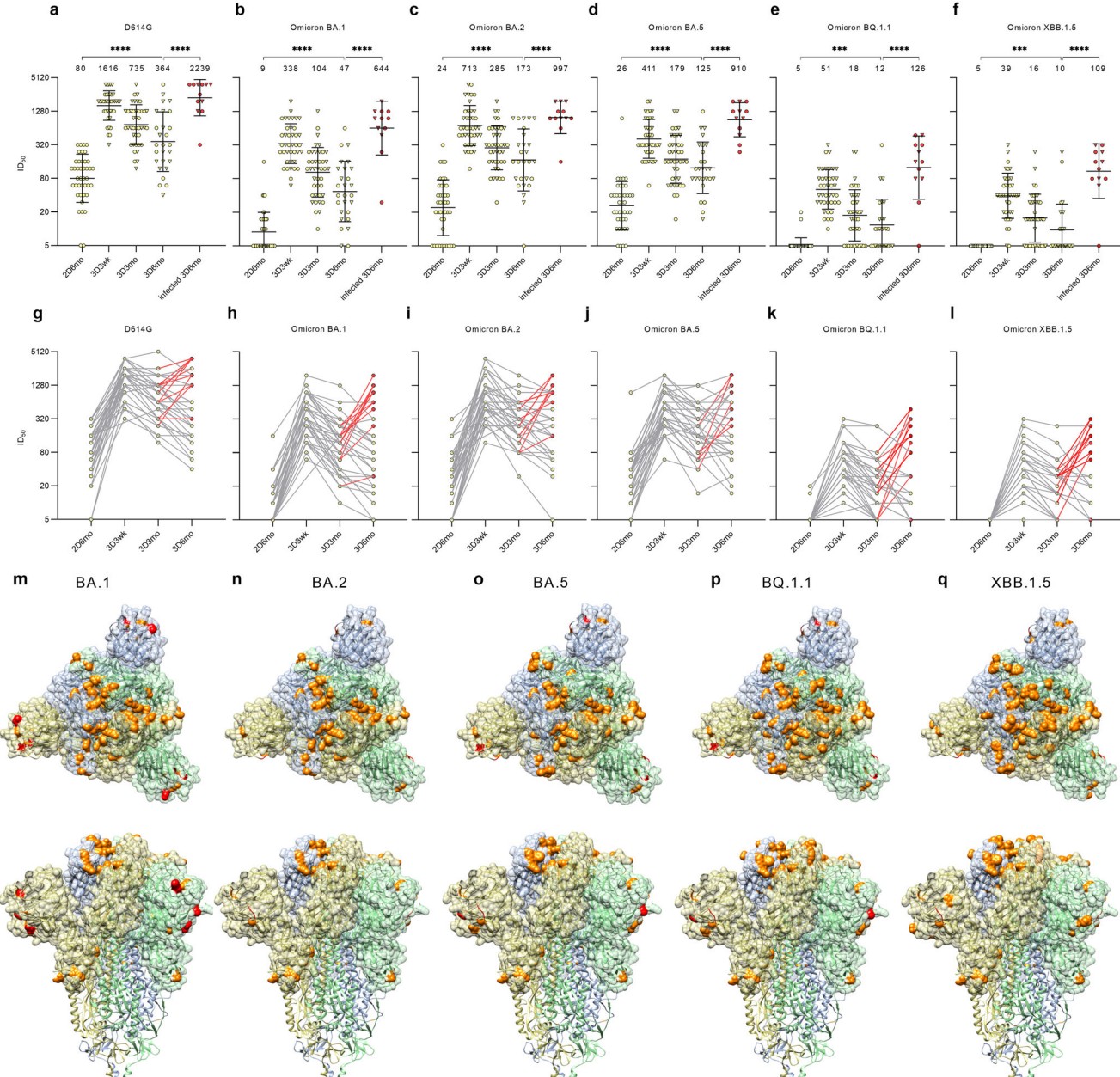

**Fig. 4 | Neutralizing antibodies of HCWs vaccinated with a short dose interval up to six months after three doses of COVID-19 mRNA vaccines.** HCWs received two doses of BNT162b2 with a three-week interval and a third dose of BNT162b2 (circle) or mRNA-1273 (triangle) eight months later. **a–f** Serum samples were collected 6 months after the second vaccine dose (2D6mo, $n = 41$), 3 weeks (3D3wk, $n = 40$), 3 months (3D3mo, $n = 41$), and 6 months (3D6mo, $n = 39$) after the third dose and analyzed with MNT for neutralizing antibodies against SARS-CoV-2 D614G and Omicron BA.1, BA.2, BA.5, BQ.1.1 (BA.5 subvariant), and XBB.1.5 (BA.2 subvariant) variants. HCWs with confirmed SARS-CoV-2 infection between three and six months after third dose ($n = 12$) were separated (red dots and triangles; infected 3D6mo). Half-maximal inhibitory dilutions ($ID_{50}$) were calculated, and titers <10 were marked as 5. Geometric mean titers for vaccine groups are indicated above each time point and shown as lines with geometric SDs. **g–l** Sequential serum samples of each individual are connected with lines. Red lines indicate where a vaccinee has had a PCR or antigen test confirmed SARS-CoV-2 infection. **m–q** Top and side views of trimeric SARS-CoV-2 spike protein structure (PDB: 7WK2) show amino acid differences of Omicron BA.1, BA.2, BA.5, BQ.1.1, and XBB.1.5 compared to Wuhan Hu-1 sequence as amino-acid substitutions (orange) and deletions (red).

indicate that the studied vaccine combinations elicit high titers of SARS-CoV-2 neutralizing antibodies against Omicron BA.1, BA.2, and BA.5 variants, while the titers against the Omicron BQ.1.1 and XBB.1.5 were yet relatively low. An infection within three months after three vaccine doses elicits high neutralizing antibody titers against all Omicron variants.

### Differences in neutralization of D614G and Omicron variants after three COVID-19 vaccine doses

The estimation of neutralization capacity of COVID-19-vaccinated individuals against different variants is central in deciding the need for further

vaccine doses. Here, we compared the neutralizing antibody titers of HCWs without breakthrough infection against D614G and Omicron variants BA.1, BA.2, BA.5, BQ.1.1, and XBB.1.5 in the order these variants emerged (Fig. 6). Regardless of the vaccine combination, the neutralizing capacity of the antibodies was 4.8–11.5x reduced 3 weeks and 3 months after the third vaccine dose when moving from D614G variant to Omicron BA.1, 2.1–4.6x increased from Omicron BA.1 to BA.2, and again 1.5–2.3x reduced from Omicron BA.2 to BA.5, and 4.9–9.9x further reduced to BQ.1.1 and to XBB.1.5. The only exception was the 2 x mRNA-1273 + BNT162b2/mRNA-1273 group, as in this group there was no significant difference

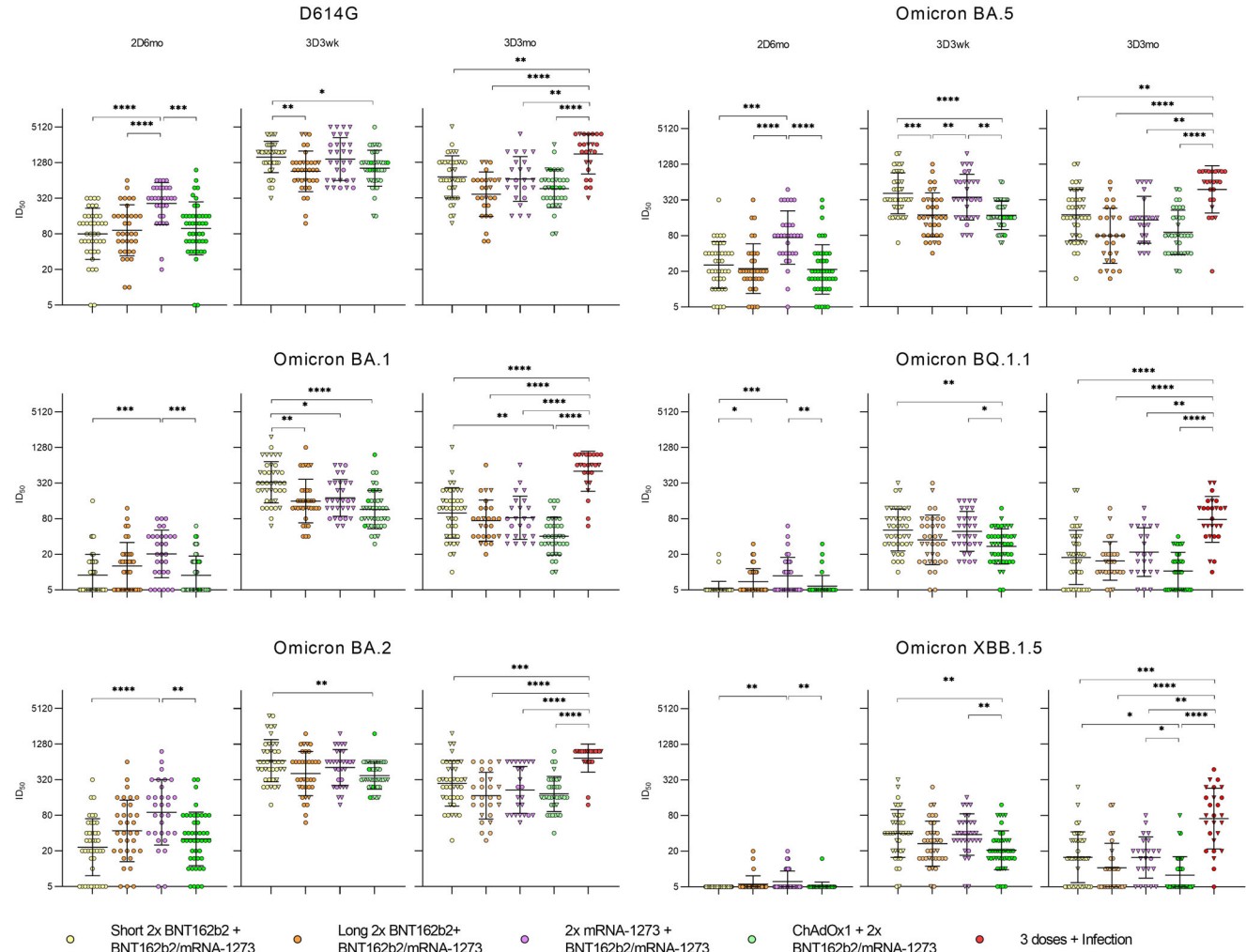

**Fig. 5 | Comparison of neutralizing antibody responses induced by different three-dose combinations of COVID-19 vaccines.** Serum samples, collected 6 months after the second dose (2D6mo), 3 weeks (3D3wk), and 3 months (3D3mo) after the third vaccine dose from HCWs who received 2x BNT162b2 with a short vaccine dose interval (n = 41, yellow circles and triangles) or 2x BNT162b2 (n = 35, orange circles and triangles), 2x mRNA-1273 (n = 31, purple circles and triangles), or ChAdOx1 + BNT162b2/mRNA-1273 (n = 45, green circles and triangles) with a long vaccine dose interval, and a third dose of BNT162b2 (circle) or mRNA-1273 (triangle) were compared for neutralizing antibody responses against D614G and Omicron variants BA.1, BA.2, BA.5, BQ.1.1, and XBB.1.5. HCWs with confirmed SARS-CoV-2 infection between the samplings at 3 weeks and three months after the third dose (3D3wk and 3D3mo) were separated (3 doses + infection, n = 24, red circles and triangles). Half-maximal inhibitory dilutions ($ID_{50}$) were calculated, and titers <10 were marked as 5. Geometric mean titers (GMTs) for each vaccine group are shown as lines with geometric SDs.

between the neutralization capacity against Omicron BA.2 and Omicron BA.5 (GMT 503 vs 359) at 3 weeks after the third dose.

The follow-up of neutralizing antibodies in HCWs who had a breakthrough infection showed that the infection boosted the titers of neutralizing antibodies 1.7–7.9x against the tested variants (Fig. 7). The fold difference between the Omicron variants was relatively similar at all time points (1.5–2.1x between BA.1 and BA.2, 1.4–1.5 between BA.2 and BA.5 etc.), while saturation of titers for D614G affected the measured GMT values (Supplementary Fig. 5). The comparable fold difference between the variants in both pre-infection and post-infection samples from vaccinees suggests that an Omicron variant infection equally enhances the existing neutralizing antibody response against various variants.

**Correlation of neutralizing antibody responses against SARS-CoV-2 D614G and five Omicron variants and with anti-S1 and anti-N IgG responses**
To analyze the correlation of the antibody responses induced by the vaccinations against the Omicron variants BA.1, BA.2, BA.5, BQ.1.1, and XBB.1.5 and the ancestral D614G variant, the neutralization efficiency of

499 serum samples from 155 vaccinees was pairwise compared between the six SARS-CoV-2 variants (Supplementary Fig. 6). Neutralizing antibody titers against the D614G variant were higher than against any of the Omicron variants. Even though the neutralization efficiency between different variants varied, they correlated well with each other (r = 0.7771–0.8766, p < 0.0001 for all pairwise comparisons).

**Congruence of Omicron BA.5 neutralizing, and S1 and N-binding antibodies**
Parallel examination of neutralizing antibody titers (for BA.5 as an example) and IgG responses for S1 and N showed similar kinetics following the third vaccine dose and breakthrough infections in both short and long vaccine interval groups (Supplementary Fig. 7). In the long vaccine interval group the breakthrough infections (n = 23) occurred closer to the third vaccine dose (between 3D3wk and 3D3mo) than in the short vaccine interval group (n = 12, between 3D3mo and 3D6mo). Of the 12 short vaccine group HCWs with a confirmed breakthrough infection 92% (11/12) had an increase in anti-N and 83% (10/12) in anti-S1 antibodies, and 83% (10/12) had at least a 4x-increase in Omicron BA.5-specific neutralizing titers. In the long-

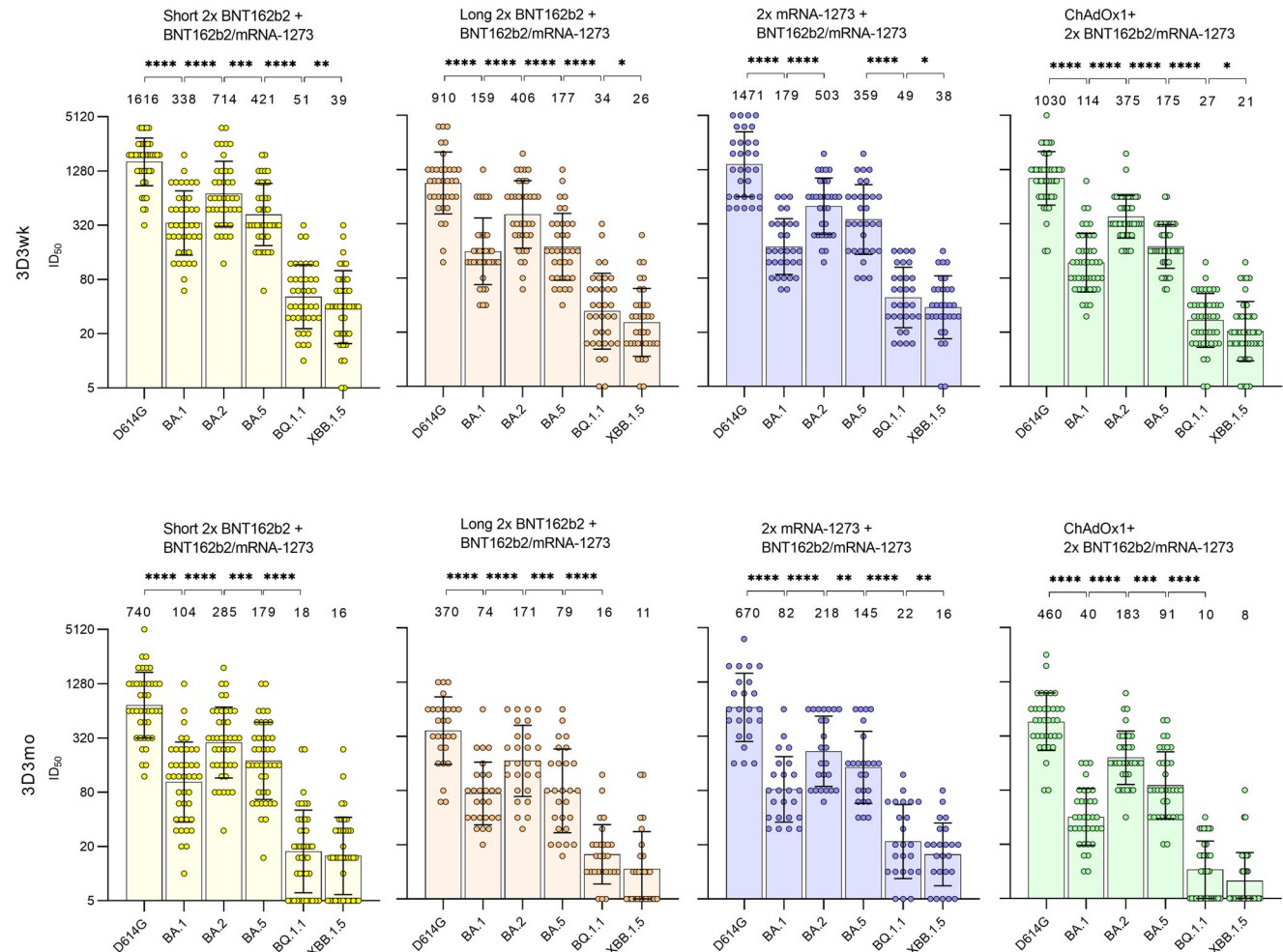

**Fig. 6 | Neutralizing antibody capacity of different COVID-19 vaccine combinations against SARS-CoV-2 D614G and different Omicron variants.** Comparison of neutralizing antibody titers against D614G and Omicron variants BA.1, BA.2, BA.5, BQ.1.1, and XBB.1.5 at 3 weeks (3D3wk) and three months (3D3mo) after the third vaccine dose of HCWs without SARS-CoV-2 infection within each vaccine combination group (2x BNT162b2 with a short, $n = 40$ at 3D3wk and at 3D3mo or a long vaccine dose interval, $n = 34$ at 3D3wk and $n = 27$ at 3D3mo; 2x mRNA-1273 $n = 31$ at 3D3wk and $n = 24$ at 3D3mo; or ChAdOx1 + BNT162b2/mRNA-1273, $n = 44$ at 3D3wk and $n = 35$ at 3D3mo). Half-maximal inhibitory dilutions ($ID_{50}$) were calculated, and titers <10 were marked as 5. Geometric mean titers (GMTs) for each vaccine group are indicated above bars and shown as lines with geometric SDs.

interval vaccine group serological evidence for a breakthrough infection was detectable in anti-N antibodies in 65% (15/23), in anti-S1 antibodies in 78% (18/23), and for Omicron BA.5 neutralizing titers in 48% (11/23) of infected HCWs. Three HCWs with a COVID-19 test-positive infection showed no increase in S1- or N-specific IgG antibodies, however, two of these had their serum sampling close (8–10 days) to the positive COVID-19 test date, and this may have been too early after the infection to detect newly formed antibodies.

## Discussion

This study assessed SARS-CoV-2-specific binding and neutralizing antibody responses in HCWs who received three doses of different COVID-19 vaccines with or without subsequent breakthrough infection. Our data showed an increasing number of serologically verified infections in the study population after the third vaccine dose. Infected vaccinees showed increases both in S1 and N-specific antibody responses as well as in the neutralizing capacity against the SARS-CoV-2 D614G variant and five different Omicron variants. More recently emerged variants, Omicron BQ.1.1 and XBB.1.5, have shown an increased ability to avoid neutralization[13–16] and our results are well in line with these observations. Regardless of the combination of three vaccine doses, the majority of the HCWs in our study had detectable levels of neutralizing antibodies also against Omicron BQ.1.1 and

XBB.1.5 directly after the third vaccine dose or after a breakthrough infection, but the number of positive samples declined relatively fast.

The decline in antibody levels after the third COVID-19 vaccine dose followed a similar kinetics to the responses seen after the first and second doses. A significant surge of breakthrough infections during the sample-taking period after the third dose coincided with the arrival of the BA.1 and BA.2 Omicron variants to Finland in late 2021 and early 2022, followed by BA.5 in spring and summer 2022[17]. The effect of these variants was especially noticeable in Finland since the previous epidemic waves had been relatively mild, and in most recruited HCWs the breakthrough infection was the first SARS-CoV-2 infection[17]. Ninety-five percent of the HCWs with a PCR or antigen detection confirmed breakthrough infection had increased SARS-CoV-2-specific antibody levels. However, in only 62% of the cases, both S1 and N-specific antibody levels increased, indicating that both antibody tests are important for a serological verification of an infection. Additionally, a recent vaccine-induced increase in antibody levels seemed to limit or mask the infection-induced increase in the antibodies. A great variation in antibody responses after SARS-CoV-2 infection has also been observed in non-vaccinated individuals, especially following symptomatically mild infections[8,18].

Pre-existing antibodies in vaccinees may restrict the infection, which can then lead to a relatively low induction of antibody responses

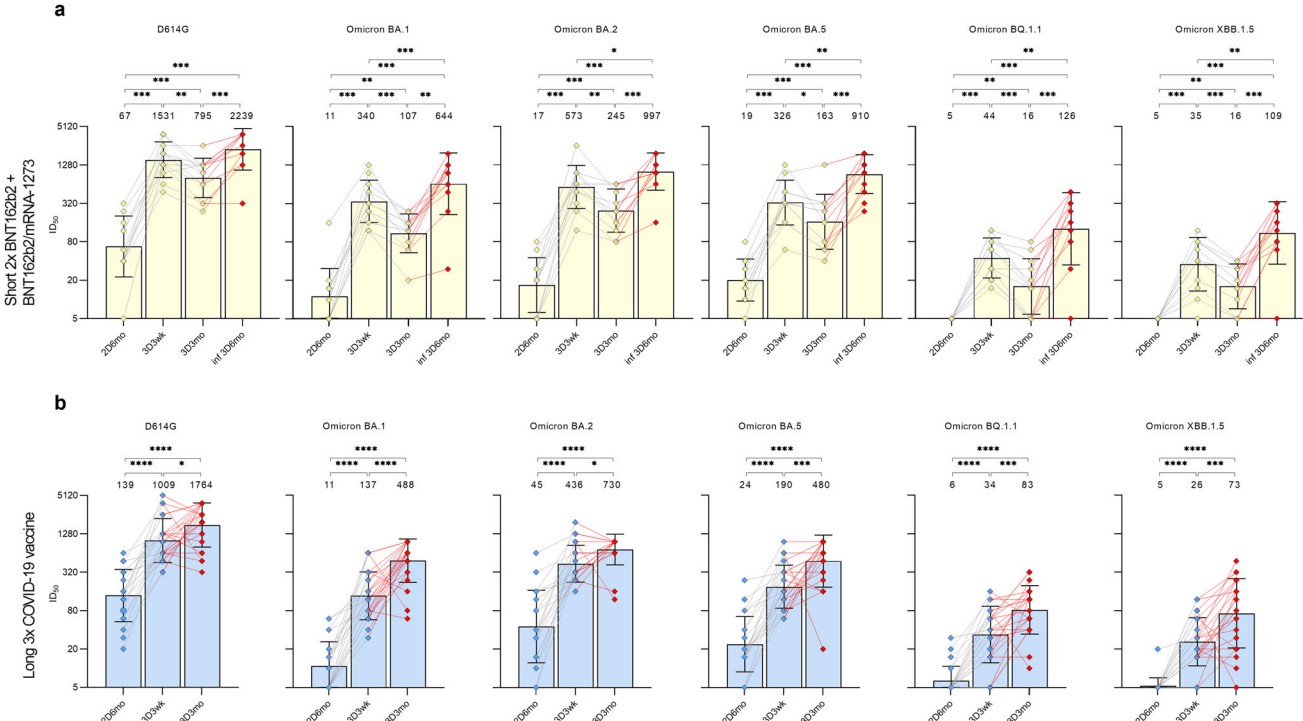

**Fig. 7 | Follow-up of neutralizing antibodies against six SARS-CoV-2 variants in HCWs with a SARS-CoV-2 infection after three vaccine doses.** Neutralizing antibodies against D614G and Omicron variants BA.1, BA.2, BA.5, BQ.1.1, and XBB.1.5 of **a** 12 HCWs with short vaccination interval and **b** 23 HCWs with long vaccination interval with a breakthrough infection were compared. Half-maximal inhibitory dilutions (ID$_{50}$) were calculated before third COVID-19 vaccine dose (2D6mo), and 3 weeks (3D3wk), 3 months (3D3mo), and 6 months (3D6mo, only for short vaccine interval group) after receiving the third COVID-19 vaccine dose. Uninfected individuals are marked with blue (long interval) or yellow (short interval) diamonds, infected individuals with red diamonds. Titers <10 were marked as 5. Titers of each vaccinee in different time points are connected with lines. Red line indicates the period of a PCR- or serology-confirmed infection. Geometric mean antibody titers with standard deviations are shown as bars and lines at each time point.

after infection. Our criteria for serologically confirmed infection were relatively strict, especially when the sample collection was not always optimal to detect an antibody rise and, at the same time, the decline in S1-specific antibodies post-vaccination and infection followed the normal kinetics. In this context, our observation of infection-related antibody increases in 14% of vaccinees with negative COVID-19 test results or no testing seems accurate but also indicates that many of the vaccinees that contracted an infection were asymptomatic or had very mild symptoms leading to no testing. Also, it is possible that the infection leads to low levels of secretion of viruses, leaving the tests negative. In unvaccinated individuals with mild COVID-19 the IgG antibodies have been shown to arise 12–14 days after SARS-CoV-2 infection[19]. Thus, the absence of detectable N-specific antibody response and non-detectable increase in S-specific (binding or neutralizing) antibodies in some vaccinees may be explained by the short interval (<14 days) between the infection and sampling.

It is generally agreed that currently neutralizing antibodies against SARS-CoV-2 is the best correlate for the protection against infection[20–22], and keeping the neutralizing titers high by revaccinations may be important at least for medical risk groups. Our analysis of the neutralizing antibodies showed that vaccinees had higher titers against Omicron BA.2 and BA.5 than against BA.1, suggesting that escaping neutralization is not the sole factor in the evolution of SARS-CoV-2. This difference in the antibody evasion capacity between Omicron BA.1, BA.2 and BA.5 seems to be associated with the number and position of amino-acid changes on the surface of the trimeric spike protein. It is noteworthy that the advantage of Omicron BQ.1.1 and XBB.1.5 over the preceding variants has been shown to arise from enhanced ACE2 binding and increased number of amino acid changes in the spike leading to further enhanced antibody evasion[16]. This advantage was observed as a much lower neutralizing capacity in the sera of

vaccinees in this study as well. Vaccinees who received a bivalent Wuhan-BA.5 vaccine as the fourth dose had substantially enhanced neutralization activity against Omicron BQ.1.1 and XBB.1.5[14]. Unfortunately, our participants had not received a fourth vaccine dose, because, in Finland this is only recommended for those at risk for severe COVID-19 infection. Future variants are likely to accumulate further amino acid changes in immunologically critical sites of the spike protein, leading to a need to update future vaccines.

The breadth of neutralizing antibodies has been shown to increase after a breakthrough infection[23]. In vaccinated HCWs with an Omicron variant breakthrough infection, we also observed an increased neutralization activity against all studied variants, which suggests that a booster response from a breakthrough infection is likely greater than that from an additional vaccine dose. Breakthrough infections by any variant boosted antibody levels, alongside neutralization capacity, to the upper interquartile range, regardless of how long it had been since the third vaccination dose. Interestingly, the neutralization capacity after three doses was less affected by the preceding vaccine combination used than the dosing interval. Similar to other studies[24], we observed a greater neutralizing antibody response with a long interval after the second vaccine dose. Notably, the third vaccine dose boosted the neutralization capacity of the short-interval group higher compared to those seen in the long-interval groups. Interestingly, the mRNA-1273 vaccine induced significantly higher S1-specific antibody levels compared to the BNT162b2, likely because of the higher mRNA content of mRNA-1273. Higher antibody levels correlate with a better neutralization capacity[25], and thus may provide better neutralizing capacity, likely providing better or longer protection against severe SARS-CoV-2 infections.

In summary, our data shows that COVID-19 vaccine-induced humoral immune response does not prevent breakthrough infections by

Omicron variants, since, in more than 50% of vaccinees, there was a microbiologically or serologically confirmed SARS-CoV-2 infection. It was also of interest that, like a recent vaccine dose, a breakthrough infection increases the neutralizing capacity against the newer Omicron variants BQ.1.1 and XBB.1.5. The main benefit of the third and likely additional vaccine doses is to boost humoral immunity rather than T cell-mediated immunity, which remains stable for longer periods of time after COVID-19 vaccination and provides protection by providing cross-recognition between variants, both of which are likely essential in preventing a severe COVID-19 disease[11,26]. In this context, our results on infection-induced antibody responses suggest that hybrid immunity should be taken into account in vaccination policy and in predicting the impact of subsequent waves of the COVID-19 epidemic, and that the development of new vaccines to combat emerging variants should be continued to reduce the number of breakthrough infections.

## Data availability

Source data, provided as Supplementary Data 1, includes all data analyzed in this paper. The SARS-CoV-2 sequence data generated in this study have been deposited in the Genbank and/or GISAID database under accession codes MW717675.1 and EPI_ISL_412971, EPI_ISL_8768822.2, OP199045 and EPI_ISL_9695067, OP199047 and EPI_ISL_13118918, OQ411064 and EPI_ISL_15762173, and OQ509907 and EPI_ISL_16526646.

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

## Acknowledgements

We thank all the HCWs for their participation in the COVID-19 vaccination studies. We gratefully acknowledge Soili Jussila, Anne Suominen, Anne-Mari Pieniniemi, and Outi Debnam for their excellent technical assistance. This study was supported by the Academy of Finland (grant number 337530 to I.J., 339512 to L.K., and 336439 and 335527 to A.K.), Jane and Aatos Erkko Foundation (grant numbers 3067-84b53 and 5360-cc2fc to I.J.), The Finnish Government Subsidy for Health Science Research (TYH2023238, to A.K.), the Finnish Medical Association (to A.K.) and the Sigrid Jusélius Foundation (to I.J. and L.K.).

## Author contributions

L.K., I.J., and P.K. designed the study. A.R., S.Ma., P.J., E.A., S.T., M.B., L.K., and P.K. performed the experiments. A.R. and P.K. analyzed the data. L.I., P.A.T., J.L., S.Mi., H.H., and A.K. performed recruitment and sample and participant data collection. M.S., A.H., P.Ö., A.I., A.P., R.N., and O.R. provided key reagents. L.K., I.J., and P.K. supervised the study. A.R., L.K., I.J., and P.K. wrote the manuscript, and all authors contributed to editing the text and approved the manuscript.

## Competing interests

The authors declare no competing interests.
