## [Peer Review File · Communications Medicine]

Reviewers' comments:

Reviewer #1 (Remarks to the Author):

The authors present a very well written and clear manuscript based on a health care study cohort (n=432) of SARS-CoV-2 vaccination and breakthrough infection in Finland. Approximately half of the cohort had a short-week interval and half had a long-week interval regimen between vaccine 1 and 2.

Serology was performed on peripheral blood to assess the effect of three mRNA spike-based vaccine combinations on antibody binding and neutralization. Firstly a timecourse of S1 and N-specific IgG responses are shown in the short-interval vaccine group which shows boosting after vaccine dose 3 and significant waning. However, infection with omicron shows a significant boost of antibody responses, bringing levels back up to dose 3. A similar effect is seen with individuals who receive the long-interval regimen.

The Authors then go on to discuss the role and correlation between of serology and PCR in defining breakthrough infection status.

The authors then discuss their finding that mRNA-1273 as a third dose provides a higher Spike antibody response than BNT162b2. The effect of age on antibody responses was investigated and no significant effect observed for this cohort.

Finally neutralizing antibody binding to omicron variants was assessed and found to be highest after 3 doses as previously reported by other studies. Breakthrough infection was reported and found to be high, in mostly an infection-naïve cohort, and induced strong antibody responses.

Specific comments or suggestions.

1. Is it necessary to split the participants by the short and long interval when investigating effect on variants post V3? Other work shows that immune responses become equivalent (Moore Med 2023) after 3rd dose for vaccine different vaccine regimens. Do the authors find the same finding here and can the authors justify the decision to keep them split for this study?
2. The data on serology versus PCR status in breakthrough infection status is confusing and the point of this part of the work isn't clear and doesn't sit that well with the flow of the results. In addition, the authors also don't discuss the possibility of false positives in PCR testing and the effect of repeated PCR testing on their findings.
3. Can the authors state what the fold change and/or mean difference is when comparing antibody responses.
4. Please discuss further the finding that mRNA-1273 as a third dose provides a higher Spike antibody response than BNT162b2 and whether the difference is biologically relevant.
5. The cohort is clearly valuable as a real-world study of SARS-CoV-2 infection and vaccination, however it would be useful if the authors made it clearer which data is novel and which is corroborating other studies.
6. There is other work that investigates similar questions around the role of breakthrough infection and in the case of previously-infected individuals the role of antigenic sin. Whilst this cohort is mostly infection naïve, this cohort can add value to this important and ongoing issue. (eg Hornsby Nat Comm 2023, in which breakthrough infection was assessed after 2 doses of vaccine in both infection naïve and previously infected individuals, in mucosal and blood and also assessed T cell

responses. And Reynolds Science 2022 which suggests antigenic sin may create a dampening of immune responses after omicron breakthrough infection.

Thank you for your work and good luck in your future studies.

Reviewer #2 (Remarks to the Author):

Reinholm et al. report a serological study in healthcare workers vaccinated with different SARS-CoV-2 vaccine combinations of BNT162b2, mRNA-1273 and ChAdOx1. Their main findings are that the different vaccine regimes induce anti-S1 and N and neutralizing antibodies, which are consistently less efficient against the most recent Omicron variants, BQ.1.1 and XBB.1.5. Breakthrough infections in HCW vaccinated three times were common and resulted in a boost in antibody responses compared to those uninfected at the same timepoint.

I enjoyed the in-depth analysis performed by the authors. However, I still have some comments and suggestions that in my opinion could help improve the presentation of the results. Please, find them below:

Introduction

- 1) There are several references for THL. Please specify on the text to which one it refers to each time.
- 2) Line 66. Please, define HCWs when first used.

Results

- 3) A limitation of the study is that most of the participants are females and some of the characteristics observed at the immune level might be different in males.
- 4) Numbers in lines 84 and 85 are not matching with those in Table 1.
- 5) Paragraph from lines 99-107 is missing a reference to the Figure 1 and its corresponding panels. This figure shows the same data in different formats to illustrate different messages. I would appreciate an explanation of the main message from each panel in the results section. Same goes for Figure 2.
- 6) In addition, in Figures 1c, 1f, 2c and 2f the lines and trends are difficult to see properly. I would suggest some of these options: to color the lines according to increase or decrease or maybe show this in a heatmap with the fold change between timepoints on the x-axis and the individuals on the y-axis.
- 7) In lines 114-116, the authors say that "The first vaccine dose induced a wide range of S1-specific antibody levels, especially in previously infected HCWs (Figure 2b-c, 115 black dots).", but to me it seems that all black dots are above the upper IQR, showing a boost of the immune response to those antigens. The levels in those previously infected does not have such a wide range.
- 8) Lines 125-127, the authors say that "Also, the N-specific antibody levels were induced by an infection, although the N-specific response was absent or low in some of the infected vaccinees (Figure 2e-g).", but checking the lines, there seems to be an increase in the following timepoint above the cutoff. Are there still some individuals that never seroconvert in any of the timepoints after the infection? Were these individuals responding to the vaccine with lower antibody levels?
- 9) In line 133, of how much was the increase? This is important because slight increases might be due to technical variations and could be the source of false positive results.
- 10) In Table 2, which timepoints were used for the calculations? The authors say it is in vaccinees

after the 3rd dose, I assumed it was either 3D3mo, 3D6mo or 3D9mo because at 3D3wk it would not be possible to distinguish the increase caused by infection or vaccine, at least for S1.

11) In Supplementary Figure 1 it should be indicated that the black dots represent the four vaccinees with a PCR positive test between third dose and three weeks after third dose (I assumed this was the case).

12) Regarding the results from Supplementary Fig 1 and 2, could it be that the boost induced by the vaccine masks or limits the increase in antibody levels, at least for S1? And after that decay, infections are able to generate another boost?

13) The section "Agreement of serology and COVID-19 PCR or antigen-test" should be summarized in my opinion, since I think it goes beyond the objectives of the paper and it is a bit lengthy.

14) In Figure 3b, were the differences observed at the first timepoint maybe due to unequal distribution of the vaccines used for the first 2 doses? Could the authors show with different colors which individuals were vaccinated with which vaccines?

15) Seeing these differences, the authors cannot state that the different vaccine schedules have similar results. They show similar kinetics but the mean levels of induced antibodies are higher (particularly at the peak response) when mRNA-1273 is used as a third dose vaccine. In addition, Figure 5 also shows statistically significant differences in the neutralizing capacity induced by each combination. This statement should be modified in the abstract.

16) The panels in Figure 4 and 5 are a bit too small to be comfortably readable.

17) In Figure 4 numbers above points should be defined in the legend.

18) In Figure 4 I would like to see as well the comparison between those infected (red dots) and the 3D3wk timepoint (peak response) to show whether infection boosts the neutralizing capacity above that reached at the peak response after vaccination.

19) In Figure 7, panel letters are missing. Also, shouldn't there be also a panel for breakthrough infections for the long group?

20) Figure 7 as it is not able to properly show what the authors state in text in lines 271-277. The authors should show the trends of each individual along the time until reaching infection and statistically compare the microneutralization titers before and after infection.

21) Do lines 274-277 refer to the "original antigenic sin" phenomenon? I suggest some rephrasing in that sentence since it was a bit difficult for me to understand what the authors wanted to express. Also, I would suggest to comment about this phenomenon in the discussion.

22) Lines 290-303 seem to belong to a different section and not to the section talking about the correlation of IgG and neutralizing antibodies. I suggest Figure 8 to be placed on supplementary material.

23) Regarding lines 301-303, the authors state that "Altogether, our data showed that although the majority of breakthrough infections were afterwards serologically detectable, a very recent administration of COVID-19 vaccine reduced the increase in anti-S1, anti-N and neutralizing antibody responses." But to state this, the authors should compare people within the same vaccination regime that got infected close to the administration of the vaccine or later in time, also taking into account variant waves. I assume the authors extract this conclusion from the fact that in the long vaccine interval group the breakthrough infections occurred closer to the third vaccine dose than in the short vaccine interval group, but it should be mentioned that the emergence of variants might be the cause of this quicker breakthrough infections. The peak antibody response in the short group was by the end of delta variant wave while the peak response in the long group was in the middle of the BA.1 and beginning of BA.2 waves. In addition, a reduced increase might be due to already high levels induced by the vaccine, therefore limiting the boost that the infection can induce.

Discussion

24) Lines 335-337, couldn't this be also due to delayed response to the vaccine? Increase in N IgG levels should allow for this distinction

25) Lines 366-367 is probably due to exposure to variants?

26) Line 368 has a typo, please remove "in".

27) Could the authors elaborate or speculate about the reasons behind the statements in lines 367-371? "Similarly to other studies (Hall et al. 2022) we observed greater neutralizing antibody response in with long interval between the first and the second vaccine dose. Notably, the group of vaccinees with short interval between first and second dose but longer interval between second and third dose had the lowest antibody levels before the third dose and the highest levels after it."

28) In the conclusions, I would mention the importance of updating the vaccines with newer variants due to the possibility of original antigenic sin and the observed fact that the existing antibodies induced by the vaccine are less efficient against the newer omicron variants.

Methods

29) In lines 415-417 the authors explain how the seropositivity cutoff was calculated. They say it was done with prevaccination samples excluding participants with prior SARS-CoV-2 infection), but how can asymptomatic or untested participants be ruled out? This could have affected the seropositivity percentages.

Reviewer #1 (Remarks to the Author):

The authors present a very well written and clear manuscript based on a health care study cohort (n=432) of SARS-CoV-2 vaccination and breakthrough infection in Finland. Approximately half of the cohort had a short-week interval and half had a long-week interval regimen between vaccine 1 and 2.

Serology was performed on peripheral blood to assess the effect of three mRNA spike-based vaccine combinations on antibody binding and neutralization. Firstly a timecourse of S1 and N-specific IgG responses are shown in the short-interval vaccine group which shows boosting after vaccine dose 3 and significant waning. However, infection with omicron shows a significant boost of antibody responses, bringing levels back up to dose 3. A similar effect is seen with individuals who receive the long-interval regimen.

The Authors then go on to discuss the role and correlation between of serology and PCR in defining breakthrough infection status.

The authors then discuss their finding that mRNA-1273 as a third dose provides a higher Spike antibody response than BNT162b2. The effect of age on antibody responses was investigated and no significant effect observed for this cohort.

Finally neutralizing antibody binding to omicron variants was assessed and found to be highest after 3 doses as previously reported by other studies. Breakthrough infection was reported and found to be high, in mostly an infection-naïve cohort, and induced strong antibody responses.

We thank the Reviewer 1 for their careful assessment and supportive comments on our manuscript.

Specific comments or suggestions.

1. Is it necessary to split the participants by the short and long interval when investigating effect on variants post V3? Other work shows that immune responses become equivalent (Moore Med 2023) after 3rd dose for vaccine different vaccine regimens. Do the authors find the same finding here and can the authors justify the decision to keep them split for this study?

As the Reviewer 1 points out, it has been shown that the immune responses reach the equivalent level after the 3rd dose regardless of the vaccine regimen. In our opinion, to complement the previous results, analyzing the vaccine regimens separately is necessary to provide a clear and detailed picture of the effects of these vaccinations after different doses and in the context of national vaccination program in Finland. Most importantly, here we detect statistically significant differences in neutralizing antibody titers between short- and long-interval groups after the 3rd vaccine dose (Figure 5 and lines 282-284). The sample size in short-interval vaccine group is larger in our study compared to for example the study of Moore et al. (Med 2023) (230 vs. 84 participants), which further strengthens the observation of the difference between short- and long-interval groups regarding the neutralizing antibodies. Thus, we feel that it is important to keep the groups separately for more thorough analysis of the immune responses. Analysis of the differences is especially informative, since in our study the groups are larger than in previous publications.

2. The data on serology versus PCR status in breakthrough infection status is confusing and the point of this part of the work isn't clear and doesn't sit that well with the flow of the results. In addition, the authors also don't discuss the possibility of false positives in PCR testing and the effect of repeated PCR testing on their findings.

Our purpose here, after presenting our serological data where we have separated breakthrough infections, is to justify our criteria for breakthrough infections. We assign as breakthrough infections PCR/antigen test positive ones and those individuals in whom we see a significant increase in antibody levels/titers between the vaccinations. For this, we want to show that our serological assay indeed recognizes those individuals who have contracted a natural infection. We have only nine vaccinees who are PCR/antigen test positive but do not show an increase in spike and/or N protein specific antibody levels after the infection (173 vs. 182), and we have discussed this (lines 402-415). To assist the interpretation of these results, we have now simplified the data presentation in Table 2 and we have revised the text accordingly.

The possibility of false positive PCR test results is very unlikely (although not impossible). PCR testing was done in the diagnostic laboratory of the hospital district where all the Covid-19 PCR testing of the whole area (pop. 600 000) has been done since the beginning of the pandemic. The laboratory is an accredited Clinical microbiology laboratory of Turku University hospital. In addition, PCR (and antigen test) results were verified by our serological test, confirming that the PCR tests were correct positives. We do, however, see individuals who show an increase in SARS-CoV-2 antibody levels between the collected samples, but who have been negative in PCR/antigen tests or from whom PCR/antigen samples have not been collected due to minimal symptoms or subclinical infection (exposure). We do see any discrepancy in our assay results, since the assays detect different variables. PCR may be positive e.g. 7-10 days while increased antibody levels retain much longer after an infection. Thus, we included an increase in antibody levels/titers between the vaccinations as an additional criteria of a breakthrough infection.

3. Can the authors state what the fold change and/or mean difference is when comparing antibody responses.

We apologize for not clearly stating this important information and we have now added numerical values to the relevant places in the text (e.g. lines 107-108, 128, 146-147, and 150-153).

4. Please discuss further the finding that mRNA-1273 as a third dose provides a higher Spike antibody response than BNT162b2 and whether the difference is biologically relevant.

This indeed is an interesting observation. As described by Steensels et al. (JAMA, 2021), the increase in the neutralization capacity is associated with higher mRNA content in the Moderna (mRNA-1273) vaccine. One mRNA-1273 vaccine dose contains more mRNA (100 ug) than BNT162b2 (30 ug), and this likely one explanation for the observed phenomenon i.e. higher antibody levels induced by mRNA-1273. The difference is likely biologically relevant to some extent, since the decay time of the antibodies is the same regardless of the vaccine, and thus the higher antibody levels induced by a vaccine result in antibodies circulating for longer time. We have discussed this now on lines 443 -

451. In addition, there is published data showing that after two doses the protective efficacy of Moderna vaccine is somewhat better. However, this type of data has to be interpreted with caution, since usually different vaccines have not been compared in the same randomized studies. Thus, we have not really emphasized the potential differences in the immunogenicity of different vaccines.

5. The cohort is clearly valuable as a real-world study of SARS-CoV-2 infection and vaccination, however it would be useful if the authors made it clearer which data is novel and which is corroborating other studies.

We agree with the Reviewer 1 that the cohort we have is extremely valuable for follow-up studies on Covid-19 infections and vaccinations. It is evident that in the Covid-19 pandemic situation, the data obtained in different countries and in different cohorts is published as soon as possible whenever the study provides new information. This leads to a situation that the new data obtained from the subsequent samples in a cohort relies also on the published data obtained from the previous samples. The situation is, however, dynamic when antibody levels decline after vaccination and new variants of concern pop up and cause global morbidity. In the revised manuscript, we have now more clearly defined the novel data on lines 77-81.

6. There is other work that investigates similar questions around the role of breakthrough infection and in the case of previously-infected individuals the role of antigenic sin. Whilst this cohort is mostly infection naïve, this cohort can add value to this important and ongoing issue. (eg Hornsby Nat Comm 2023, in which breakthrough infection was assessed after 2 doses of vaccine in both infection naïve and previously infected individuals, in mucosal and blood and also assessed T cell responses. And Reynolds Science 2022 which suggests antigenic sin may create a dampening of immune responses after omicron breakthrough infection.

Original antigenic sin is an important phenomenon, and we agree this cohort would be valuable in this context as well. However, to fully understand the putative effect of the antigenic sin in Covid-19 vaccine- and infection-induced antibody responses, detailed epitope-focused studies would be needed. These experiments are beyond the scope of this current study, although those are likely addressed in our forthcoming studies.

Thank you for your work and good luck in your future studies.

We thank the Reviewer 1 for the valuable comments that helped us to improve our manuscript.

Reviewer #2 (Remarks to the Author):

Reinholm et al. report a serological study in healthcare workers vaccinated with different SARS-CoV-2 vaccine combinations of BNT162b2, mRNA-1273 and ChAdOx1. Their main findings are that the different vaccine regimes induce anti-S1 and N and neutralizing antibodies, which are consistently less efficient against the most recent Omicron variants, BQ.1.1 and XBB.1.5. Breakthrough infections in HCW vaccinated three times were common and resulted in a boost in antibody responses compared to those uninfected at the same timepoint.

I enjoyed the in-depth analysis performed by the authors. However, I still have some comments and suggestions that in my opinion could help improve the presentation of the results. Please, find them below:

We thank the Reviewer 2 for careful assessment and positive comments on our manuscript. We have addressed the comments and suggestions in detail below.

Introduction

1) There are several references for THL. Please specify on the text to which one it refers to each time.

We have now revised the references for THL that denotes Finnish Institute for Health and Welfare and refers to webpages indicated in reference numbers 2, 6 and 17.

2) Line 66. Please, define HCWs when first used.

Thank you for noticing this, we have now defined the abbreviation HCW properly in line 74 of the revised manuscript.

Results

3) A limitation of the study is that most of the participants are females and some of the characteristics observed at the immune level might be different in males.

Our cohort consisted of voluntary participants of healthcare workers, a profession in Finland dominated by women, which biased our cohort towards females. It is possible, that there are differences in immune responses between females and males, however, in our previous studies we have shown that after two Covid-19 vaccine doses the differences in vaccine induced antibody levels between the sexes are not significant (Jalkanen et al Nat Com 2022).

4) Numbers in lines 84 and 85 are not matching with those in Table 1.

We thank the Reviewer for pointing out this mistake and we have now corrected the numbers to revised lines 95 and 96.

5) Paragraph from lines 99-107 is missing a reference to the Figure 1 and its corresponding panels. This figure shows the same data in different formats to illustrate different messages. I would appreciate an explanation of the main message from each panel in the results section. Same goes for Figure 2.

We separated the data into multiple panels to show the overall responses, follow-up of responses for each vaccinee and responses based on the infection status. This was done in order to make the

data presentation more reader-friendly, since we had a vast number of participants (n=430). We have now revised the text to clarify the message in each panel.

6) In addition, in Figures 1c, 1f, 2c and 2f the lines and trends are difficult to see properly. I would suggest some of these options: to color the lines according to increase or decrease or maybe show this in a heatmap with the fold change between timepoints on the x-axis and the individuals on the y-axis.

We agree that these figures may be a bit busy. However, the purpose of these figures is to provide an overview of the data and trends of the uninfected and infected individuals at a general level. To show the trends in detail for each individual is not our purpose. Nevertheless, as suggested by the Reviewer 2, we tested different ways of presenting the data of figure 1c (see below) and conclude that the observation of trends remains challenging even if the data is presented in other ways. We do, however, feel that other colours (blue, option 2) or heatmap (option 3; we lose the absolute values) do not improve the readability of the data. Thus, we would like to leave our original figures to the manuscript.

Explanatory figure 1. Panels from left to right: 1) original figure 1c, 2) increases in antibody levels in red and decreases in blue, 3) heatmap showing the fold change between adjacent samples (white space presents no sample and fold changes of >5 are marked as 5).

Explanatory figure 2. Anti-S1 and anti-N responses of the COVID-19 test positive (shortened to PCR-pos) vaccinees with (n=144, a-b) and without (n=38, c-d) indication of infection in anti-N antibody levels. Vaccine-induced anti-S1 antibody levels at three weeks after the third dose (3D3wk) of the two groups were also compared (e). Geometric mean titers are shown as numbers and bars with standard deviations.

7) In lines 114-116, the authors say that “The first vaccine dose induced a wide range of S1-specific antibody levels, especially in previously infected HCWs (Figure 2b-c, 115 black dots).”, but to me it seems that all black dots are above the upper IQR, showing a boost of the immune response to those antigens. The levels in those previously infected does not have such a wide range.

We have revised lines 137-139 to convey the correct statement.

8) Lines 125-127, the authors say that “Also, the N-specific antibody levels were induced by an infection, although the N-specific response was absent or low in some of the infected vaccinees (Figure 2e-g).”, but checking the lines, there seems to be an increase in the following timepoint above the cutoff. Are there still some individuals that never seroconvert in any of the timepoints after the infection? Were these individuals responding to the vaccine with lower antibody levels?

There were indeed vaccinees who showed low or no antibody response to a PCR/antigen verified infection (please see the attached explanatory figure 2). We set the criteria for serological indication of an infection according to the cut-off value (an increase greater than cut-off). Thus a few vaccinees whose anti-N levels increased (even above cut-off) but who also had slightly elevated basal anti-N response, failed to fill the criteria for anti-N positivity. The majority of these vaccinees had a diagnostic increase in anti-S1 antibodies, however, nine of the infected vaccinees did not seroconvert during the follow-up. These observations may indicate that in some vaccinated individuals the infection is so mild that an increase in anti-N antibodies is not induced in significant levels.

9) In line 133, of how much was the increase? This is important because slight increases might be due to technical variations and could be the source of false positive results.

This has been made clearer. We have now stated in the revised text (line 163-164) that the increase is significant. We have taken into account the technical variation by calculating the coefficient of variation for the controls with 140 duplicate repeats, which gave the anti-S1 EIA CV% (percentage of coefficient of variation) of 8.42 and anti-N EIA CV% of 5.7. We have now added this information to line 497-501. The technical variation in the test is very low.

10) In Table 2, which timepoints were used for the calculations? The authors say it is in vaccinees after the 3rd dose, I assumed it was either 3D3mo, 3D6mo or 3D9mo because at 3D3wk it would not be possible to distinguish the increase caused by infection or vaccine, at least for S1.

We have now revised Table 2 and it includes now only the key numbers for our comparison. The Reviewer is correct regarding the anti-S1 response and 3D3wk time point. We wanted to include all of the infections after the third vaccine dose and thus we included the interval between the third dose and 3D3wk based on the data on anti-N antibodies coupled with COVID-19 PCR/antigen tests.

11) In Supplementary Figure 1 it should be indicated that the black dots represent the four vaccinees with a PCR positive test between third dose and three weeks after third dose (I assumed this was the case).

We thank the Reviewer 2 for noticing this. We have now added the description for the black dots in the Supplementary figures 1 and 2.

12) Regarding the results from Supplementary Fig 1 and 2, could it be that the boost induced by the vaccine masks or limits the increase in antibody levels, at least for S1? And after that decay, infections are able to generate another boost?

Reviewer 2 is correct in this, and indeed this is what we also think has happened. A sentence regarding this was added to the manuscript, lines 180-181, and 398-399.

13) The section “Agreement of serology and COVID-19 PCR or antigen-test” should be summarized in my opinion, since I think it goes beyond the objectives of the paper and it is a bit lengthy.

We have now summarized the data in a new Table 2 and have also extensively revised the paragraph. In our opinion the remaining part of this paragraph is relevant to include in this manuscript, as it reinforces the validity of our serological tests, and provides insight into the kinetics of hybrid immunity, especially post- third dose infection. Our data also emphasizes the value of antibody detection methods in microbial diagnostics and seroprevalence studies.

14) In Figure 3b, were the differences observed at the first timepoint maybe due to unequal distribution of the vaccines used for the first 2 doses? Could the authors show with different colors which individuals were vaccinated with which vaccines?

We agree with the Reviewer 2 and believe this is the case. With the long interval mRNA-1273 vaccinees, previous two vaccine doses were also mRNA-1273, and since mRNA-1273 has a higher mRNA concentration and has been shown to induce higher antibody levels (Wang et al., JAMA. 2022), this is likely the cause of the difference between BNT and mRNA-1273 groups. We have now revised Figure 3 and marked by colors the individuals based on their first 2 vaccines. Also, we added a notion regarding the pre third dose vaccine groups to the results section, lines 215-216.

15) Seeing these differences, the authors cannot state that the different vaccine schedules have similar results. They show similar kinetics but the mean levels of induced antibodies are higher (particularly at the peak response) when mRNA-1273 is used as a third dose vaccine. In addition, Figure 5 also shows statistically significant differences in the neutralizing capacity induced by each combination. This statement should be modified in the abstract.

We have now modified the Abstract to describe the kinetics and neutralization capacity (line 37).

16) The panels in Figure 4 and 5 are a bit too small to be comfortably readable.

We have now increased the size of the panels to make them more reader friendly.

17) In Figure 4 numbers above points should be defined in the legend.

We have now added the definitions to the figure legend.

18) In Figure 4 I would like to see as well the comparison between those infected (red dots) and the 3D3wk timepoint (peak response) to show whether infection boosts the neutralizing capacity above that reached at the peak response after vaccination.

We have now illustrated these peak differences in the new Figure 7.

19) In Figure 7, panel letters are missing. Also, shouldn't there be also a panel for breakthrough infections for the long group?

We have now added panel letters and corrected the naming of the panels (headline of the infected long group was incorrect). The old figure 7 is now Supplementary Figure 5 in our revised manuscript. Data in the new Figure 7 is now shown as a follow-up of each vaccinee.

20) Figure 7 as it is not able to properly show what the authors state in text in lines 271-277. The authors should show the trends of each individual along the time until reaching infection and statistically compare the microneutralization titers before and after infection.

We have now modified this figure to show the trends of each individual. The new figure 7 supports better the description of the observations in this paragraph of the text. The old Figure 7 is now Supplementary Figure 5 in our revised manuscript.

21) Do lines 274-277 refer to the "original antigenic sin" phenomenon? I suggest some rephrasing in that sentence since it was a bit difficult for me to understand what the authors wanted to express. Also, I would suggest to comment about this phenomenon in the discussion.

The lines referred by the reviewer have now been revised and made clearer to convey the correct statement, and are now on revised lines (326-329).

22) Lines 290-303 seem to belong to a different section and not to the section talking about the correlation of IgG and neutralizing antibodies. I suggest Figure 8 to be placed on supplementary material.

We have now separated lines 290-303 to be a separate section named "Congruence of Omicron BA.5 neutralizing, and S1 and N binding antibodies", now line 357-372, and moved Figure 8 to the supplementary material as a Supplementary Figure 7. Furthermore, we have added more explanatory text to the paragraph.

23) Regarding lines 301-303, the authors state that “Altogether, our data showed that although the majority of breakthrough infections were afterwards serologically detectable, a very recent administration of COVID-19 vaccine reduced the increase in anti-S1, anti-N and neutralizing antibody responses.” But to state this, the authors should compare people within the same vaccination regime that got infected close to the administration of the vaccine or later in time, also taking into account variant waves. I assume the authors extract this conclusion from the fact that in the long vaccine interval group the breakthrough infections occurred closer to the third vaccine dose than in the short vaccine interval group, but it should be mentioned that the emergence of variants might be the cause of this quicker breakthrough infections. The peak antibody response in the short group was by the end of delta variant wave while the peak response in the long group was in the middle of the BA.1 and beginning of BA.2 waves. In addition, a reduced increase might be due to already high levels induced by the vaccine, therefore limiting the boost that the infection can induce.

We thank the Reviewer 2 for pointing this out, and we have now removed the incorrect statement, and added discussion on this to revised lines 398-399.

Discussion

24) Lines 335-337, couldn't this be also due to delayed response to the vaccine? Increase in N IgG levels should allow for this distinction

Delayed response to vaccination is possible, however, we see that very unlikely, since the antibodies induced by vaccination should be detectable on average 2-3 weeks after the administration, with or without previous infection, according to our previous research and literature (e.g. Wei et al., Nat Microbiol, 2021).

25) Lines 366-367 is probably due to exposure to variants?

While this could be a possible reason, this would still affect neutralization and dosing interval in a similar manner. We can see this happen in the Figure 6. when comparing short/long interval groups and the effect of different vaccine combination while having the variants trends as context for changes in overall neutralization capacity.

26) Line 368 has a typo, please remove “in”.

The typo has now been removed.

27) Could the authors elaborate or speculate about the reasons behind the statements in lines 367-371? “Similarly to other studies (Hall et al. 2022) we observed greater neutralizing antibody response in with long interval between the first and the second vaccine dose. Notably, the group of vaccinees with short interval between first and second dose but longer interval between second and third dose had the lowest antibody levels before the third dose and the highest levels after it.”

Briefly, our interpretation of the data was that longer interval between the second and the third vaccine dose seemed to be advantageous regarding neutralization capacity. Previously, we and others have shown this to be true for the interval between the first and the second vaccine dose. Here we observed that in the short interval group, which had on average 9 months between the second and the third dose, had higher neutralizing capacity after the third dose than what was seen in the long interval group, which on an average had only 6 months between the second and the third dose. We have now clarified this message in the manuscript.

28) In the conclusions, I would mention the importance of updating the vaccines with newer variants due to the possibility of original antigenic sin and the observed fact that the existing antibodies induced by the vaccine are less efficient against the newer omicron variants.

We do acknowledge the potential effect of the original antigenic sin, but since we have not studied that phenomenon in detail here, we refrain from commenting on that in our manuscript. Nevertheless, we have now added a sentence to the conclusion to bring up the importance of variant-specific vaccine development (lines 463-464). This has already been decided by FDA and EMA (preference for XBB.1.5).

Methods

29) In lines 415-417 the authors explain how the seropositivity cutoff was calculated. They say it was done with prevaccination samples excluding participants with prior SARS-CoV-2 infection), but how can asymptomatic or untested participants be ruled out? This could have affected the seropositivity percentages.

The reviewer is correct here. We could rule out only the participants who had a PCR-confirmed infection. Therefore, our percentages of seropositivity may be marginally lower than the true number. To minimize the effect of the antibody responses from asymptomatic or untested individuals into the calculation of the cut-offs we conducted the calculations in two rounds. First we calculated the average of EIA unit values from all of the prevaccination samples (excluding those with a confirmed infection) and added three times the standard deviation. This gave us the preliminary cut-off. Then the values exceeding the preliminary cut-off (mean + 3 SD units) were excluded before calculating the final cut-offs. The calculation strategy was clarified into the manuscript.

We thank the Reviewer 2 for the insightful comments that helped us to improve our manuscript.

We feel that we have now adequately responded to all concerns raised by the reviewers and when appropriate we have also modified the manuscript text accordingly. We hope that after these modifications you find the paper suitable for publication in Communications Medicine.

Sincerely,

Arttu Reinholm, M.Sc., Ilkka Julkunen, M.D., Ph.D., Professor, Pekka Kolehmainen, Ph.D.

Reviewers' comments:

Reviewer #1 (Remarks to the Author):

Thank you for the revised manuscript. I feel it is much improved and the authors have addressed all my comments thoughtfully.

A couple of things are outstanding.

1. There are a few instances where units for the binding assay are missed in the text (I believe these were EIA units according to the methods). If a conversion is possible can WHO BAU/ml also be displayed?

2. Thank you for explaining clearly in the methods which data has been previously published, it is understandable and encouraged for a cohort such that data is published when it is not "complete". However, to prevent confusion and to aid transparency please can you state clearly in the relevant figure legends which data displayed has been previously published (with references). I would advise if the authors make their data publicly available that all DOIs are linked back to the original dataset for good data curation.

Thank you

Reviewer #2 (Remarks to the Author):

I appreciate the response of the authors and the changes they have made to the manuscript addressing the comments of both reviewers. I believe the concerns I raised were all answered or properly justified if changes were not made.

However, I still have one concern, and this is the definition of serological detection of breakthrough infection. The authors have now specified that this is based on levels of antibodies being above the cutoff. However, I believe this definition does not take into account those participants that might already have levels above the cutoff but nonetheless may have significant increases in antibody levels due to an infection. How did the authors account for this?

Thank you

Reviewers' comments:

Reviewer #1 (Remarks to the Author):

Thank you for the revised manuscript. I feel it is much improved and the authors have addressed all my comments thoughtfully.

A couple of things are outstanding.

1. There are a few instances where units for the binding assay are missed in the text (I believe these were EIA units according to the methods). If a conversion is possible can WHO BAU/ml also be displayed?

We have added the EIA units to the revised text for clarity. Conversion of the OD values to an international standard would have been highly desirable, but the shortage and difficulty in acquiring WHO International Antibody Standard reagents unfortunately made it too difficult for us to obtain the standard. The reliability and standardization of our in-house assay has been previously described in detail (Jalkanen et al., JID, 2021) and proved in multiple publications of our group.

2. Thank you for explaining clearly in the methods which data has been previously published, it is understandable and encouraged for a cohort such that data is published when it is not "complete". However, to prevent confusion and to aid transparency please can you state clearly in the relevant figure legends which data displayed has been previously published (with references). I would advise if the authors make their data publicly available that all DOIs are linked back to the original dataset for good data curation.

Thank you

Relevant references have been added to the revised figure legends. We thank Reviewer #1 for their additional comments and hope that these revisions are now acceptable.

Reviewer #2 (Remarks to the Author):

I appreciate the response of the authors and the changes they have made to the manuscript addressing the comments of both reviewers. I believe the concerns I raised were all answered or properly justified if changes were not made.

However, I still have one concern, and this is the definition of serological detection of breakthrough infection. The authors have now specified that this is based on levels of

antibodies being above the cutoff. However, I believe this definition does not take into account those participants that might already have levels above the cutoff but nonetheless may have significant increases in antibody levels due to an infection. How did the authors account for this?

Thank you

In most cases, the breakthrough infections were clearly observable due to a substantial fold increase in EIA units/neutralizing dilutions post-breakthrough. However, in cases with a more subtle increase, the cut-off was used to determine the difference between pre- and post-breakthrough serum samples. If the difference was more than the cutoff, for example 4.8 EIA units in the case of S1, it was determined that a breakthrough infection had occurred. To avoid confusion regarding the criteria for infection we now modified the text part on lines 426-429 describing the criteria.

We thank Reviewer 2. for their comment, and hope we have answered this concern adequately.

Sincerely,

Arttu Reinholm, M.Sc.

Pekka Kolehmainen, Ph.D., senior scientist

Ilkka Julkunen, M.D., Ph.D., Professor

Reviewers' comments:

Reviewer #1 (Remarks to the Author):

Thank you - all comments fully addressed.
best of luck

Reviewer #2 (Remarks to the Author):

Thank you for your answer.

When I read the rebuttal letter, I understand that in most cases the breakthrough infections were detected by measuring the fold change between timepoints in samples for which the antibody levels were already above the cutoff. However, in the modified manuscript I cannot find this information reflected in the methods section.

It would be necessary to indicate which fold change was considered when detecting breakthrough infections in samples that were already above the cutoff.

If the authors address this point, I have no further comments and I would accept the manuscript for publication.

Thank you.

Reviewers' comments:

Reviewer #2 (Remarks to the Author):

When I read the rebuttal letter, I understand that in most cases the breakthrough infections were detected by measuring the fold change between timepoints in samples for which the antibody levels were already above the cutoff. However, in the modified manuscript I cannot find this information reflected in the methods section.

It would be necessary to indicate which fold change was considered when detecting breakthrough infections in samples that were already above the cutoff.

We apologize for the confusion caused by our answer to Reviewers comments. In our study the serological indication of an infection was assessed as described in the manuscript, i.e. only by comparing the anti-S1 and anti-N antibody levels (measured by EIA) of consecutive serum samples. For anti-spike antibodies an increase of 4.8 EIA units (cut-off for anti-S1 antibodies) or more between consecutive samples was considered to indicate an infection (excluding the time points around vaccinations where detection of infection-induced anti-S1 antibody level increase was not possible). For anti-N antibodies an increase indicating an infection was 8.8 EIA units. The cut-offs were defined as the mean EIA unit values of seronegative specimens + 3 SD units and thus an increase above the cut-off was significant (cut-off for anti-N antibodies).

In the previous rebuttal letter the notion on fold changes referred to changes in neutralizing antibody titers, which match well with the changes in anti-S1 and anti-N antibodies as shown in supplementary figure 7. However, fold changes in neutralizing antibodies were not taken in consideration in determination of breakthrough infections.

We hope our answer has clarified the issue.

Sincerely,

Arttu Reinholm, M.Sc.

Pekka Kolehmainen, Ph.D., senior scientist

Ilkka Julkunen, M.D., Ph.D., Professor

REVIEWERS' COMMENTS:

Reviewer #2 (Remarks to the Author):

Thank you for the answer.

The authors have clarified my question.

I have no further comments.

All the best.